# Dynamic Changes in Colonic Structure and Protein Expression Suggest Regulatory Mechanisms of Colonic Barrier Function in Torpor–Arousal Cycles of the Daurian Ground Squirrel

**DOI:** 10.3390/ijms23169026

**Published:** 2022-08-12

**Authors:** Weilan Miao, Yuting Han, Yingyu Yang, Ziwei Hao, Ning An, Jiayu Chen, Ziwen Zhang, Xuli Gao, Kenneth B. Storey, Hui Chang, Shiwei Wang

**Affiliations:** 1Shaanxi Key Laboratory for Animal Conservation, Northwest University, Xi’an 710069, China; 2Key Laboratory of Resource Biology and Biotechnology in Western China (College of Life Sciences, Northwest University), Ministry of Education, 229# North Taibai Road, Xi’an 710069, China; 3Department of Biology, Carleton University, Ottawa, ON K1S 5B6, Canada

**Keywords:** torpor–arousal, colonic mucosal, tight junction, permeability, barrier function

## Abstract

Background: Both pathological conditions and hibernation can affect the barrier function of small intestine mucosa. However, the effect of hibernation on the barrier function of colonic mucosa remains unclear. Methods: We investigated morphological changes in colonic mucosa, the concentrations of specific proteins and molecules, and the enzymatic activity of diamine oxidase (DAO), in serum and colonic tissue; the expression of tight junction proteins and mucin, and the changes in inflammatory, farnesoid X receptor (FXR)–small heterodimer partner (SHP), and apoptosis-related molecules that could play a role in gut permeability changes in Daurian ground squirrels in summer active (SA), late torpor (LT), and interbout arousal (IBA) periods. Results: The results show that hibernation reduced the thickness of the colonic mucosa and the depth of the crypt, decreased the number of goblet cells (GCs), and damaged the structure of some microvilli. The concentrations of proteins and molecules, and the enzymatic activity of DAO, were all increased in the serum and colon, and the localization of tight junction proteins and mucin in the colonic mucosa were altered (compensatory response). Although the ground squirrels ate during the interbout arousal period, the changes remained similar to the response to torpor. Inflammation, apoptosis–anti-apoptosis, and FXR–SHP signaling may be involved in the possible changes in intestinal gut permeability during the torpor–arousal cycle in Daurian ground squirrels. In addition, periodic interbout arousal may play an inflammation-correcting role during the long hibernation season of Daurian ground squirrels.

## 1. Introduction

Mammalian hibernation is a striking adaptation used by some species during periods of relative food scarcity and harsh environmental conditions [1]. Hibernation is associated with long periods of metabolic depression, and fasting for some species, such as the European hamster (*Cricetus cricetus*) [2], the thirteen-lined ground squirrel (*Ictidomys tridecemlineatus*) [3], and the Arctic ground squirrel (*Spermophilus undulatus*) [4]. Daurian ground squirrels (*Spermophilus dauricus*) are small rodents; as a typical fat-storing hibernating mammal, it is similar to other small hibernating mammals in that its metabolic rate decreases significantly (1–5%), body temperature (Tb) decreases to near ambient temperature (as low as 1–5 °C), and blood flow, heart rate, and oxygen consumption, decrease significantly during torpor. However, this state is not constant throughout the hibernation period, but is interrupted by several interbout arousals, accompanied by rapid recovery of many of these physiological functions, including a return to normal body temperature (35–38 °C), oxygen consumption, and metabolic rate [5], during an arousal period of no more than 24 h.

Daurian ground squirrels enter an intermittent fasting and water deprivation state throughout the whole hibernation season, which means no food and water during torpor, but they eat and drink during interbout arousal (see the details in the Animals and Groups subsection of the Materials and Methods section), unlike the other hibernating mammals that have been reported. Thus, we speculate that this unique hibernation habit of the Daurian ground squirrel may lead to changes in its physiological indices during the torpor–arousal cycle that differ from the results of previous studies. Moreover, previous studies have focused primarily on the small intestine and the first portion of the large intestine (cecum), whereas, to date, there is very little information on seasonal and metabolic state-specific changes distal to the cecum. Furthermore, studies on the effects of hibernation on the gastrointestinal tract have also not been carried out previously with Daurian ground squirrels. Therefore, the present study was proposed to detect seasonal and metabolic state-specific changes in the colon of Daurian ground squirrels, which could provide some new information to our current understanding of how hibernation in mammals affects the structure and potential function of the gastrointestinal tract.

Intestinal epithelial cells form a physical barrier of tight, adherent junctions and bridging grains [6]. The paracellular permeability of these barriers is maintained by different proteins, such as claudin, occludin, MUC2, E-cadherin, and zonula occludens (ZO) [7]. These connections open and close frequently in response to various stimuli. For example, 2 days of weaning made crypts of the small intestine of piglets (26 days of age) shallow [8], whereas the permeability of the jejunum and ileum of chickens (38-day-old Ross-308 males) increased after fasting for 4.5, 9 and 19.5 h [9]. Electron microscopic imaging revealed an abnormal cellular morphology and enlarged subepithelial spaces of ischemic enterocytes of rat jejunum, subcellular necrotic and organelle destruction (including mitochondrial swelling, cytoplasmic vacuolation, microvilli in microsomal shapes and plasma membrane disintegration) [10]. Ischemia–reperfusion (IR) in the rat small intestine (60 min of ischemia and 60 min of reperfusion) resulted in increases in plasma intestinal fatty-acid-binding protein (I-FABP), endotoxin (ET), L-lactate (D-LA), and mRNA levels of the pro-inflammatory cytokines interleukin-6 (IL-6) and interleukin-1-β (IL-1-β) [11]. In addition, water deprivation and weaning stress significantly reduced the mRNA expression levels of occludin, claudin-1, and ZO-1 in the jejunum and ileum of piglets [12]. After 6 h of ischemia or hypoxia–reperfusion, the gene and protein expression levels of claudin-1, occludin, and ZO-1 were decreased in mouse small intestinal epithelium, and staining of the three proteins showed typical rearrangements and significant disruptions, such as diffuse and discontinuous distribution [13]. Deletion or mutation of E-cadherin resulted in a reduction of small intestinal epithelial cells, abnormal tight junction complex attachment, and impaired barrier function in mice [14]. In addition, mucins secreted by goblet cells (GCs) are also considered to be the first line of defense against any pathology and inflammation in the intestine [15], and the number of GCs, as well as the protein and mRNA expression level of MUC2, are reduced in pathological conditions, when the intestinal barrier is dysfunctional and intestinal permeability is increased [16]. Thus, we wondered if the structure and function of the intestinal mucosal barrier of hibernating mammals can be maintained after they have undergone hibernation? What are the mechanisms of maintenance or regulation? The investigation of these questions will not only provide new insights into the physiological strategies of hibernating animals to cope with the adverse environment, but also provide new ideas and targets for the development of various intestinal disease control measures in non-hibernators (including humans).

There are a few reports in the literature that the depth of the small intestinal crypts was significantly reduced in hibernating greater mouse-eared bats (*Myotis myotis*) compared with the active phase [17]. The small intestinal epithelium of European hamsters [2] and thirteen-lined ground squirrels [3] showed atrophy during food deprivation in hibernation. Relative to summer, hibernation (early winter and late winter) reduced the cecal crypt length (the depth of crypt) of the thirteen-lined ground squirrels, and the number of GCs per crypt was reduced in the late winter ground squirrels compared to the other groups. In addition, the protein expression of MUC2 in cecum tissues varied seasonally, with its expression being higher in early winter and spring compared to summer and late winter, but its expression in late winter animals did not differ from other seasonal groups [18]. The amount of mucus observed in large intestine GCs was lower when hibernating Arctic ground squirrels were in torpor from the amount observed in the warm hibernators and in active squirrels [4]. Evidence from thirteen-lined ground squirrels suggested there was higher gut permeability during the hibernation season compared to summer squirrels. Transepithelial ion conductance measurements suggest that passive ion flow across the small intestine is elevated during hibernation [3,19] and the luminal to blood movement of the macromolecular marker FITC-dextran is also elevated, which also indicates total gut permeability increases in hibernation. Expression of the tight junction protein occludin in ground squirrel cecum including its phosphorylated form is greater in hibernating than in summer squirrels, as is the localization of occludin in the apical intercellular spaces where it can regulate tight junction permeability [20]. Hence, from these studies it is evident that there are a series of changes that occur in the gastrointestinal tract of hibernating mammals that fast during winter, some of which may be deleterious (e.g., increased permeability), and others that may be compensatory adaptations to the permeability change (e.g., occluding expression and localization) [20]. However, the above hibernating mammals fast during the entire hibernation season. Unlike these animals, Daurian ground squirrels eat during the interbout arousal period. Unfortunately, to date, no study has reported relevant changes in the gut of hibernating mammals that eat during the interbout arousal period. Therefore, the present study is of great interest to investigate the changes in colonic mucosal morphology and proteins or molecules that may be related to their barrier function during the torpor–arousal cycle and their possible regulatory mechanisms, using Daurian ground squirrels as a study subject. Interestingly, a previous study found that the levels of tumor necrosis factor α (TNF-α) and Interleukin 10 (IL-10) in the small intestinal mucosa of thirteen-lined ground squirrels were significantly increased during hibernation, especially during interbout arousal period [21]. Therefore, we speculated that the high expression level of IL-10 may reduce the potentially deleterious effects of pro-inflammatory cytokines released during interbout arousal and thus exert a protective effect on the intestinal barrier during hibernation. Furthermore, as one of the nuclear receptors, farnesoid X receptor (FXR), whose activation was shown to attenuate ischemia–reperfusion injury in the rat small intestine [11]. Moreover, FXR expressed in activated immune cells and counter-regulates the expression of inflammatory cytokines [22,23]. In addition, small heterodimer partner (SHP), as a downstream factor of FXR, can also synergistically regulate intestinal barrier homeostasis with FXR after its activation or transcriptional upregulation [24]. Furthermore, in mammals, the stem cells of the intestinal crypts normally experience a low level of apoptosis to eliminate mutated or defective cells and control overall cell numbers [25]; however, apoptosis increases in response to a variety of pathologies and stresses, including lack of luminal nutrition [26,27,28], ischemia–reperfusion [29], and oxidative stress [30]. Therefore, in the present study, changes in inflammatory factors, FXR–SHP, and apoptotic signaling were examined during the torpor–arousal cycle of ground squirrels.

In summary, we hypothesize that eating during interbout arousal may alter the morphology of the colonic mucosa, the concentrations of certain proteins and molecules in the serum, and the enzymatic activities of DAO in the colon and serum, as well as the localization of tight junction proteins and mucin, in Daurian ground squirrels. Inflammatory factors, and the FXR–SHP and apoptosis–anti-apoptosis signaling pathways, may be involved in regulating these changes; however, knowledge of such regulatory mechanisms is limited. For this reason, we conducted the following study.

## 2. Results

### 2.1. HE Staining of Colonic Mucosa

Hematoxylin–eosin (HE) staining was performed on the colonic mucosa of ground squirrels, and observations from the light microscope showed that the colonic mucosa of summer active (SA), late torpor (LT), and interbout arousal (IBA) groups of ground squirrels presented multiple differences in surface epithelium, crypt, and lamina propria. The subepithelial basolateral membrane of the surface epithelium was thick, whereas the epithelial basolateral membrane of the crypt was thin. The surface epithelium was mainly composed of columnar epithelial cells, occasionally interspersed with individual goblet cells (GCs), while a large number of GCs were present in the crypt. The lamina propria between the crypts was covered with columnar epithelial cells. Compared with the SA group, partially closed, round, and independent crypts appeared in the lower part of the primary crypt in LT and IBA groups; that is, the crypt morphology was significantly changed (Figure 1A). Statistical analysis showed that the mucosal thickness in the LT and IBA groups was significantly decreased by 24.2 % and 19.3% (*p* < 0.01), respectively, compared with the SA group. However, compared with the LT group, the mucosal thickness in the IBA group was significantly increased by 19.3% (*p* < 0.01) (Figure 1B). The depth of the crypt was also significantly reduced by 22.9% in the LT group compared with the SA group (*p* < 0.05), whereas there was no significant change in the IBA group (only a tendency to reduce, *p* = 0.089) compared with the LT group (Figure 1C).

### 2.2. AB–PAS Staining of Colonic Mucosa

The colonic mucosal tissues of ground squirrels were stained with Alcian blue/periodic acid-Schiff (AB–PAS) and observed under a light microscope to assess the morphological changes of GCs. The results show that the GCs of the SA, LT, and IBA groups were regularly dispersed on the mucosal surface and crypt areas with high electron density (Figure 2A). The counting results show that the number of GCs in the colonic mucosa was significantly decreased by 50.3% and 15.7% (*p* < 0.01) in the LT and IBA groups, respectively, compared to the SA group. However, the number of GCs was significantly increased by 69.7% (*p* < 0.01) in the IBA group compared with the LT group (Figure 2B).

### 2.3. Ultrastructure of Colonic Mucosal Epithelial Cells

Transmission electron microscopy (TEM) was used to observe the colonic mucosal epithelial cells of ground squirrels, and the results show that the morphology of the colonic mucosal epithelial cells of the LT and IBA groups were very similar, and that the epithelium of the surface facing the colonic lumen was mainly composed of tall columnar cells, namely absorption cells (ACs). However, the morphology of microvilli (Mv) varied greatly between different ACs or in different fields of view, as shown by the fact that some epithelia were covered with structurally intact, long, regular, and dense Mv (Figure 3B,D), and some epithelia were covered with Mv that were structurally disrupted, shortened in length, or even disintegrated into microsomes (Figure 3A,C). Of course, the disruption of Mv referred to in this study could also be the result of the ultramicrotome cutting the Mv at a certain angle, resulting in the inevitable loss of the Mv’s intact structure. However, we are inclined to believe that it is the colonic Mv of Daurian ground squirrels themselves that are disrupted during hibernation (LT and IBA groups), because previous studies have also obtained results consistent with ours [31]. In addition, we also observed that the morphology of other subcells and organelles in the colonic mucosal epithelial cells of the LT and IBA groups were normal, including an elongated nucleus in the center of epithelial cells, mitochondria with crista-like structures, dense cytoplasmic electrons, and a tight cell–matrix gap. (Figure 3A,C). High magnification electron microscopy also revealed that the actin filaments in the Mv were rooted in the terminal network (TW), and the cell junctions between adjacent cells, including tight junctions, adhesive bands, bridging granules, and gap junctions, were closely connected to the terminal network.

### 2.4. Ultrastructure of Goblet Cells in Colonic Mucosa

The GCs of colonic mucosa were observed by transmission electron microscopy. The results show that the GCs in the LT group were almost open (Figure 4A,B); that is, there was a sign of secreting mucus particles. In the IBA group, GCs were present in both closed and open configurations (Figure 4C,D).

### 2.5. Ultrastructure of Apoptotic Cells in the Colonic Mucosa

The apoptotic cells of the colonic mucosa of ground squirrels were observed by transmission electron microscopy. As shown in Figure 5, the colonocytes in the LT and IBA groups exhibited abnormal morphology, which showed that the apoptotic epithelial cells changed from a normal high columnar shape to an irregular shape, including inverted triangles. Some of the Mv covering the epithelium of apoptotic cells were deformed or even disintegrated to create vacuoles or microsomes containing cytoplasmic components, cellular granules, and apoptotic vesicles (Figure 5A,F). These partially exuded from the cytoplasm into the lumen, but some of the Mv were structurally intact, long, regular, and densely arranged (Figure 5B–D). In addition, colonocytes in the LT and IBA groups also exhibited enlarged subepithelial spaces (Figure 5A,E), disrupted organelles in individual cells (Figure 5C,D), and apoptotic bodies in the cytoplasm (Figure 5C,F). Nuclei also changed from a normal elongated central nucleus to an irregular shape, and the condensation and margination of nuclear chromatin, swollen and gathered mitochondria with disrupted or blurred cristae, and vacuolated cell fragments also appeared in the periphery of the nucleus and in the cytoplasm (Figure 5).

### 2.6. Detection of Intestinal Mucosal Barrier Permeability and Integrity Indicators

Compared with the SA group, the levels of zonulin, intestinal fatty-acid-binding protein (I-FABP), endotoxin (ET) and D-lactic acid (D-LA) proteins in serum, and the enzyme activity of diamine oxidase (DAO) in colonic tissues and serum, were all significantly increased by 44.1%, 73.7%, 45.1%, 65.6%, 76.8%, and 77% (*p* < 0.01), respectively, in LT ground squirrels. Furthermore, the levels of zonulin, I-FABP, ET, and D-LA proteins in serum, and the enzyme activity of DAO in colonic tissues and serum, were all significantly increased by 22.7% (*p* < 0.01), 62.6% (*p* < 0.01), 19.9% (*p* < 0.01), 54.9% (*p* < 0.05), 54.8% (*p* < 0.01), and 62.2% (*p* < 0.01), respectively, in IBA ground squirrels. However, compared with the LT group, the concentrations of zonulin and ET in serum, and the enzyme activity of DAO in colon tissue, were significantly decreased by 14.9%, 17.3% and 12.4% (*p* < 0.01), respectively, in the IBA group. In contrast, there were no significant differences in the concentrations of I-FABP and D-LA, or in the enzyme activity of DAO, in serum between the LT and IBA groups (Figure 6).

### 2.7. Distribution and Expression of Proteins Associated with Tight Junctions in the Colonic Mucosa

Based on the results from immunofluorescence, our analyses show that there was no significant difference in the spatial distribution of occludin, E-cadherin and ZO-1 on the colonic mucosal surface among SA, LT, and IBA groups; all exhibited strong localization in the apical membranes in the upper part of tight junctions between adjacent epithelial cells (close to the apical surface) (Figure 7). In addition, the immunoblotting results reveal that the protein expression level of occludin did not change significantly among the three groups. In contrast, the protein expression level of E-cadherin was significantly reduced by 28% in the LT group compared with the SA group (*p* < 0.05), but there was no change in the IBA group compared to the SA and LT groups (Figure 8).

Regarding the spatial localization of occludin, E-cadherin and ZO-1 in colonic mucosa, the immunohistochemical (Figure 9A) results were consistent with immunofluorescence results.And the immunohistochemical results also show that the distributions of claudin-1 on the colonic mucosal surface of ground squirrels was not significantly different among SA, LT, and IBA groups. In contrast, the expression of claudin-2 and MUC2 proteins in the colonic mucosa was higher in the upper part of the crypt in the SA group, but in the LT and IBA groups, the expression of both proteins were higher in the lower part of the crypt (Figure 9B). For immunohistochemical staining, we conducted negative control experiments to avoid false positives, and the results show that neither secondary antibodies or DAB interfered with the expression of the positive proteins during the experiment; thus, the results we obtained are plausible (Figure 9C). In addition, the results of the ELISA show that the protein concentrations of ZO-1 were significantly decreased by 38.3% and 15.8% (*p* < 0.01) in LT and IBA groups, respectively, compared with the SA group, whereas the protein concentrations of ZO-1 were significantly increased by 36.5% (*p* < 0.01) in the IBA group compared with the LT group. Similarly, the protein concentrations of claudin-1 were significantly reduced by 19.4% and 31.8% (*p* < 0.01), respectively, in the LT and IBA groups compared with the SA group, and were also significantly reduced by 15.4% (*p* < 0.05) in the IBA group compared with the LT group. However, compared with SA group, the protein concentrations of claudin-2 were significantly increased by 115.7% and 90.9% (*p* < 0.01), respectively, in the LT and IBA groups, but there was no significant difference in the concentrations of claudin-2 between LT and IBA groups. Moreover, compared with the SA group, the protein concentrations of MUC2 in LT and IBA groups were significantly increased by 118.3% and 63.6% (*p* < 0.01), respectively, whereas they were significantly decreased by 25.1% (*p* < 0.05) in IBA group compared with the LT group (Figure 10).

### 2.8. The Concentration of Inflammatory Factors in Serum and Colon

Compared with the SA group, the levels of pro-inflammatory factors TNF-α, IL-1β, and IL-6 in serum of the LT group of ground squirrels were significantly increased by 83.4%, 35%, and 106.6% (*p* < 0.01), respectively, whereas the above indicators in colon tissue of the LT group were also significantly increased by 132.2%, 44.2% and 67.3% (*p* < 0.01), respectively. Furthermore, compared with the SA group, the levels of TNF-α and IL-6 in the serum were significantly increased by 98.6% and 49.5% (*p* < 0.01) in the IBA group, respectively, and the level of IL-1β was not significantly different between IBA and SA groups, whereas the levels of TNF-α, IL-1β, and IL-6 were significantly increased by 58.8%, 30.9%, and 45.6% (*p* < 0.01) in the colon tissue of the IBA group, respectively. However, compared with the LT group, the levels of IL-1β and IL-6 in the serum of the IBA group were significantly decreased by 16.4% (*p* < 0.05) and 27.6% (*p* < 0.01), respectively, and only the TNF-α levels were not different between IBA and LT groups. In contrast, compared with LT group, only the concentration of TNF-α was significantly decreased by 31.6% (*p* < 0.01) in the colon tissue of the IBA group, whereas the concentrations of IL-1β and IL-6 were not changed (Figure 11A–F).

In addition, IL-10, as an inflammatory and immunosuppressive factor, was significantly decreased by 22.8% and 40.4% (*p* < 0.01), respectively, in serum and colonic tissues in the LT group compared with SA group, whereas IL-10 was significantly increased by 23.2% (*p* < 0.01) in the IBA group. However, there was no significant difference in the concentrations of IL-10 between the IBA and SA groups in the colon tissue. Compared with the LT group, the concentrations of IL-10 were significantly increased by 59.6% and 50.6% (*p* < 0.01) in serum and colon tissue of the IBA group, respectively (Figure 11G,H).

The concentrations of CRP, a suggestive or discriminatory indicator of bacterial (not viral) infection, were significantly increased by 79.7% and 54.3% (*p* < 0.01), respectively, in serum and colonic tissues in the LT group compared with the SA group. Serum CRP was also significantly increased by 67.3% (*p* < 0.01) and 25.4% (*p* < 0.05) in serum and colonic tissues in the IBA group, respectively, compared with SA group. However, there was no change in the concentrations of CRP in serum of the IBA group compared with the LT group, whereas CRP in the colonic tissue was significantly decreased by 18.7% (*p* < 0.01) (Figure 11I,J).

### 2.9. Protein Expression Levels of FXR and SHP in the Colon

The results show that the protein expression levels of FXR and SHP were both increased significantly in the colonic tissues in LT group by 38.9% (*p* < 0.01) and 74% (*p* < 0.05), respectively. By contrast, the protein expression levels of FXR and SHP in the IBA group did not change significantly compared to the SA group. However, the protein expression levels of FXR and SHP were significantly decreased by 37.9% (*p* < 0.01) and 43.7% (*p* < 0.05), respectively, in the IBA group compared with the LT group (Figure 12).

### 2.10. Protein Expression Levels of Bcl-2 and Bax in the Colon

Bcl-2 is an anti-apoptotic protein. The protein expression levels of Bcl-2 were significantly increased by 105.9% and 145.2% (*p* < 0.01), respectively, in the colon in both LT and IBA groups, compared to the SA group, whereas they showed no change in the colon of the IBA group compared with LT group. Furthermore, the pro-apoptotic protein, Bax, showed no expression level change in the colon in the LT group compared with the SA group, although there was a tendency to increase (*p* = 0.746). However, the protein expression level of Bax was significantly increased by 84.4% (*p* < 0.01) in the IBA group compared with the SA group, and also showed a significant increase of 76.2% (*p* < 0.01) in the IBA group compared with LT group. In addition, the Bcl-2/Bax ratio was significantly higher by 98% (*p* < 0.01) in the colon of the LT group, but was unchanged in the IBA group, compared with the SA group. Moreover, the Bcl-2/Bax ratio was significantly lower by 44% (*p* < 0.01) in the colon of the IBA group compared with the LT group (Figure 13).

## 3. Discussion

The intestinal epithelial barrier is the first line of defense against various intestinal endotoxins, allergens, and pathogens [32]. Hibernation is a physiological adaptation of mammals in response to low environmental temperatures and limited food availability. For those species that do not eat during the hibernation season, there may be a nutritional deficiency in the intestinal lumen during hibernation, which increases the permeability of the intestinal epithelium and makes the body more permeable to macromolecules during the hibernation season [20]. However, Daurian ground squirrels eat during interbout arousal, which may keep the intestine from being nutrient deficient throughout the hibernation season, and thus, intestinal permeability may not be at risk of elevated intestinal permeability during hibernation.

Firstly, we observed that there was a significant change in the morphology of the crypt in the colonic mucosa. One possibility for this to occur is that nutrient sources in the colon were limited after the ground squirrels entered hibernation and stopped feeding. In contrast, it has been shown that when the goblet cells reached the top of the crypt, most of them leave the birth and death system of the colonic epithelium by shedding into the lumen or undergoing apoptosis. That is, the crypt cells meet the nutritional requirements of low metabolism during hibernation in an apoptotic renewal manner [33], and the amino acids and phospholipids formed by the hydrolysis of the apoptotic cells can later provide nutrients to the organism. In addition, statistical analysis showed that the thickness of the colonic mucosa of ground squirrels was significantly reduced during hibernation (LT and IBA groups) (Figure 2B). It has also been shown that the jejunal mucosa undergoes atrophy (i.e., villi are shorter and crypt length is reduced) during hibernation in thirteen-lined ground squirrels [19]. In response to our study, we suggest that hibernation alters the gut microbiota, and allows the thickness of the colonic mucosa layer to be reduced during hibernation. It has been shown that winter fasting eliminates a major source of degradable substrates that support gut microbial growth, e.g., hibernation raises the relative abundance of Bacteroidetes and Verrucomicrobia in the cecum contents of thirteen-lined ground squirrels, while decreasing the relative abundance of Firmicutes), and Bacteroidetes and Verrucomicrobia contain species that can survive on host-derived substrates (e.g., mucins) [34]. In addition, the depth of crypt in the colonic mucosa of Daurian ground squirrels was also significantly reduced during hibernation (LT and IBA groups) (Figure 2). The crypt is located in the intestinal tract of the animal, and it increases the area of the intestinal surface, thus enhancing the absorptive function of the intestine. The depth of the crypt in the colonic mucosa of the ground squirrels was reduced after entering hibernation, which may be due to the reduction of intestinal absorption and secretion functions, as a result of winter fasting. Significantly, other studies have obtained results consistent with the present study, e.g., the depths of crypts in the duodenum, jejunum, and ileum of hibernating greater mouse-eared bats (*Myotis myotis*) were significantly reduced compared to those of active bats [17]. The authors suggested that the reduction in the depth of crypt may lead to lower cell proliferation and migration rates, which may be a way to reduce the cost of maintaining the intestine while hibernating. Our group agrees with the above proposal. Interestingly, the above results (thickness of colonic mucosa and the depth of crypt) were increased during the interbout arousal period compared to the LT group. In conclusion, we suggest that for those species that do not eat during the hibernation season, prolonged fasting during hibernation may not favor the rapid recovery of the intestinal mucosa during the interbout arousal period. In contrast, as displayed by Daurian ground squirrels, the intestinal mucosa may be well adapted to intermittent fasting, and this may be due to the resumption of cell renewal of the intestinal epithelium during IBA (3–5 days), which has been demonstrated in other hibernators [35]; it is this rapid renewal that allows the intestine to respond quickly to various external factors during hibernation.

Secondly, the results of AB–PAS staining show that GCs were found not only on the surface epithelium of the colonic mucosa, but throughout the crypt regions as well, and that there was high electron density in the SA, LT, and IBA groups of ground squirrels (Figure 3A). Furthermore, the number of GCs in the colonic mucosa of ground squirrels decreased significantly during hibernation (LT and IBA groups), but increased in the IBA group compared to LT group (Figure 3B). Moreover, the present study, using immunohistochemical and ELISA assays, reveals that the immunoreactivity and concentration of MUC2 significantly increased during hibernation (LT and IBA groups) compared to SA group, while it was significantly decreased in the IBA group compared to LT group (Figure 10B and Figure 11D). In contrast, the protein expression of MUC2 in cecum tissues of thirteen-lined ground squirrels varied seasonally, with its expression being higher in early winter and spring compared to summer and late winter, but its expression in late winter animals did not differ from other seasonal groups [18]. MUC2 is the main protein in GCs, and is secreted to form a mucus layer, which in turn protects the cells lining the intestinal lumen against pathogens, toxins, and gastrointestinal enzymes [36,37], and has an important role in maintaining mucosal homeostasis [38] and enhancing the mucosal barrier [39]. Under pathological and mucosal inflammatory conditions, infection leads to a decrease in mucus production and MUC2 protein expression, which directly affects the morphological and functional activity of GCs [40,41]. Physiological parameters (e.g., temperature) are reduced during hibernation [42]; nonetheless, cold stimulation can act directly on the goblet cells and promote the large-scale secretion of mucus or MUC2 [43]. This may explain why the number of GCs decreased, but the expression of MUC2 increased, during hibernation.

Transmission electron microscopic observations of epithelial cells show that the tight junction network, and the structures of the organelles and subcellular organelles, were present in the colonic mucosa. However, some microvilli disintegrated into microsomes during hibernation. It is known that microvilli increase the surface area of cells, which facilitates the absorptive function of the intestine. The microvilli disintegrate to form microsomal vesicles containing cytoplasmic components, cellular granules, and apoptotic vesicles, that further disintegrate to provide nutrients, such as amino acids, to the organism [44]. Therefore, we suggest that intact tight junctions in the colonic mucosa of Daurian ground squirrels during hibernation may counteract the increase in colonic epithelial permeability that may occur during hibernation, and may also provide morphological support or compensatory mechanisms for the overall maintenance of colonic mucosal barrier function. In addition, our observations of the colonic mucosal epithelial cells of Daurian ground squirrels using transmission electron microscope reveal, for the first time, that the colonocytes of ground squirrels exhibit typical signs of apoptosis during hibernation (LT and IBA groups). However, small intestinal epithelial cells of summer and hibernating thirteen-lined ground squirrels have been examined using the TUNEL staining method. TUNEL staining of the nucleus increased with hibernation, but the staining was diffuse and not accompanied by chromatin condensation and apoptotic vesicles [45]. Moreover, maintaining the integrity of the epithelial barrier by regulating the rate of apoptosis is also considered to be crucial for homeostasis in the intestine, and only if the rate of apoptosis matches the rate of cell renewal can the structural integrity of the intestine and intestinal physiological homeostasis be maintained [46].

To date, several blood markers have been shown to measure intestinal permeability in a variety of pathological conditions, such as, zonulin [47], I-FABP [48], ET [49], D-LA [50], and DAO [51]. Our study, using enzyme-linked immunosorbent assays, shows that the levels of zonulin, I-FABP, ET, and D-LA in the serum, and the activity of DAO in serum and colonic tissues, were significantly increased during hibernation (LT and IBA groups). However, compared to the LT group, the levels of zonulin and ET in serum and the activity of DAO in colonic tissue were significantly decreased in the IBA group; these changes in proteins, molecules, and enzymes may indicate an increase in colonic permeability during hibernation in Daurian ground squirrels (Figure 7). For those species that do not feed during the hibernation season, there may be a nutritional deficiency in the intestinal lumen during hibernation, which increases the permeability of the intestinal epithelium and makes the body more permeable to macromolecules during hibernation [20]. However, Daurian ground squirrels feed during interbout arousal, which may keep the intestine from becoming nutritionally deficient throughout hibernation, and consequently, there may also be no risk of increased intestinal permeability. This has also been shown in previous studies, i.e., intestinal permeability to macromolecular markers was increased in hibernating thirteen-lined ground squirrels during hibernation as assessed by ex vivo permeability tests. It was suggested that the altered intestinal permeability may be due to the changes in the expression of tight junction-related proteins that regulate ion, macromolecular, and paracellular channels [19]. Indeed, the results of the present study show that protein expression levels of the tight junction-related proteins E-cadherin, ZO-1, and claudin-1 were significantly reduced in the colonic mucosa of ground squirrels during hibernation (LT and IBA groups). Therefore, we speculate that the possible increased colonic permeability of Daurian ground squirrels during torpor may be associated with the downregulation of colonic mucosal tight junction-related proteins.

However, the exact mechanism of regulating or maintaining the intestinal barrier in hibernating animals is still unknown. Studies have found increased immunohistochemical staining for occludin in the small intestine and cecum of hibernating thirteen-lined ground squirrels, with particularly strong staining in apical intercellular spaces, and this result was supported by immunoblotting for hyperphosphorylated occludin in the cecum contents [20]. Occludin, one of the intestinal tight junction proteins, regulates the size and selectivity of the intestinal epithelial barrier [52]. It was suggested that the increase in occludin expression during hibernation may be a compensatory response, thus limiting the extent to which intestinal permeability increases, thereby rapidly maintaining homeostasis during the long hibernation period [30]. Although these results seem counterintuitive, they suggest that upregulation of occludin is supported when intestinal permeability is increased, such as under stressful conditions [53,54]. By contrast, our study shows that the mucosal localization and protein expression levels of claudin-2 and MUC2 were significantly upregulated during hibernation (LT and IBA groups). We suggest that the mucosal localization and protein expression levels of claudin-2 and MUC2 in the colonic mucosa of Daurian ground squirrels during torpor–arousal may play a compensatory response to the seemingly detrimental changes in colonic mucosal thickness, shallow crypt depth, reduced number of goblet cells, and possible increased colonic permeability during the torpor–arousal period. Furthermore, it has also been documented that the expression levels of TNF-α and IL-10 increase in the small intestinal mucosa of hibernating thirteen-lined ground squirrels, especially upon arousal from torpor during IBA [21]. In contrast, TNF-α production was sustained during hibernation. In the intestine, levels of both TNF-α and IL-10 rose two- to fourfold in hibernating thirteen-lined ground squirrels, although TNF-α levels fluctuated throughout the torpor–arousal cycle, reaching a peak during IBA [55]. TNF-α can disrupt the intestinal epithelial barrier by inducing apoptosis of epithelial cells [56,57,58]. IL-10 is a potent anti-inflammatory cytokine that inhibits the production and reverses the effects of several inflammatory mediators, including TNF-α [59,60]. Moreover, IL-10 may also reduce mucosal damage by inhibiting epithelial cell apoptosis [61,62,63]. In contrast, our results show that, compared to the SA group, the expression of pro-inflammatory factors as well as CRP, an indicator of bacterial infection, were significantly increased in serum and colonic tissues of ground squirrels in the LT group, while the expression of anti-inflammatory factors was significantly decreased. However, the expression of pro-inflammatory factors as well as CRP in the serum and colonic tissues of the IBA group of ground squirrels were significantly decreased, while the expression of anti-inflammatory factors was significantly increased (Figure 11). Therefore, we believe that the reason for the different changes in the aforementioned inflammatory factors may be due to inconsistencies in the study animals, study conditions, or study tissues. In view of our study, we speculate that the interbout arousal period may play a corrective role throughout the hibernation season, thus preventing the animals from overexpressing pro-inflammatory factors and causing local or systemic inflammatory responses after re-entering hibernation.

Moreover, recent animal and clinical studies have shown that dysfunction of the intestinal FXR signaling pathway also plays a key molecular switching role in intestinal ischemia–reperfusion injury (IRI), driving a loss in intestinal barrier integrity, which in turn triggers bacterial translocation, the release of inflammatory cytokines, and death of the individual [11]. Furthermore, SHP, as a downstream factor of FXR, can also synergistically regulate intestinal barrier homeostasis with FXR after its activation or transcriptional upregulation [24]. The results of our experiment show that the protein expression levels of both FXR and SHP were increased in the colonic tissues of the LT group of ground squirrels; however, the protein expression levels of both were significantly decreased again in the IBA group (Figure 12). In conclusion, we speculate that there may be a natural resistance property or mechanism to ischemia–reperfusion injury in the colon of ground squirrels during later torpor, in which FXR and SHP may play important roles.

Finally, to reveal whether apoptosis affects the colonic mucosal barrier function in hibernating ground squirrels, we investigated the expression of apoptosis-related proteins (Bcl-2 and Bax) in the colon. The results show that the protein expression level of Bcl-2 significantly increased during hibernation (LT and IBA groups), while the protein expression level of Bax only tended to increase in LT, but significantly increased in the IBA group. Meanwhile, the ratio of Bcl-2/Bax was significantly increased in the LT group compared with the SA group, while the ratio of Bcl-2/Bax was significantly decreased in IBA group compared with the LT group (Figure 13). Another study reported similar findings during an investigation of the expression of apoptosis pathway-related proteins in the jejunal mucosa of summer versus hibernating thirteen-lined ground squirrels. The authors found that anti-apoptotic Bcl-xL protein was increased 12-fold and pro-apoptotic Bax protein was increased twofold in the jejunal mucosa of hibernating ground squirrels, whereas no significant change in the protein expression level of Bcl-2 occurred, but the expression of phosphorylated Bcl-2 increased [45]. Therefore, we tentatively suggest that the anti-apoptotic signaling pathway is active in the colon of ground squirrels during the late torpor period, which may promote the survival of intestinal cells in a pro-oxidative, pro-apoptotic environment. Subsequently, during the interbout arousal period, the animals woke and began to initiate apoptotic signaling pathways, which maintained the function of the intestinal epithelium, and the continuous cell renewal and tissue homeostasis. However, the mechanism that initiates this apoptotic signaling pathway is not well understood. It may be caused by (a) increased expression levels of pro-inflammatory factors in the serum of the organism, (b) by decreased expression levels of tight junction-related proteins in the colonic mucosa, or (c) by various complex changes in the organism that together regulate and initiate the apoptotic signaling pathway. These may act to remove cells or organelles that were naturally aged, abnormally damaged, or unwanted by the organism during the later torpor period, thus playing an important role in biological and/or organismal evolution, internal environmental homeostasis, and the development of multiple systems, which remain to be further investigated.

There are several other limitations of the present study. Firstly, colonic permeability was not directly examined. Secondly, no other auxiliary methods were used in this study to label or identify apoptosis and the types of apoptosis. Finally, we only focused on the changes in mucosal morphology and barrier function in the colonic segment of ground squirrels, and did not consider the changes of morphology and barrier function in the intestinal mucosa of different segments of the gastrointestinal tract. Nevertheless, this study provides multiple new insights into colonic adaptations for hibernation that will lead to multiple new studies to more fully understand the adaptive changes at work to alter and adapt intestinal tissues to support long-term torpor in hibernating mammals. 

## 4. Materials and Methods

### 4.1. Animals and Groups

Eighteen Daurian ground squirrels (*Spermophilus dauricuss*) were captured in June and mid-September from their breeding base in Dali County, Weinan City, Shaanxi Province, China. All procedures were approved by the Laboratory Animal Management Committee of the Ministry of Health of the People’s Republic of China (Approval Number: SL-2012-42). All ground squirrels were kept in a single cage with a plastic feeding cage size of 20 cm × 25 cm × 40 cm. Daurian ground squirrels captured in June (summer active (SA)) were housed in an animal house with a temperature range of 18–25 °C and fed with standard rodent chow, water, and peanuts. After 1 week of acclimatization, the ground squirrels were weighed and samples were taken. Daurian ground squirrels captured in September were also kept in an animal house at 18–25 °C until hibernation, and fed with standard rodent chow, water, and peanuts. By late November, the ground squirrels gradually entered hibernation, at which time we transferred them to a dark hibernation house at 4–6 °C. The body temperatures of ground squirrels were monitored and recorded daily at 9 am and 9 pm using a visual thermometer with thermal imaging (Fluke, VT04, Everett, Washington, DC, USA). The hibernation period of Daurian ground squirrels was from late November to early March of the following year, with an average of 93.95 days; during this period, the animals undergo several interbout arousals. The average length of torpor in hibernation bouts was 7.44 days, and the average length of arousal between bouts was 1.36 days. The number of torpor days accounted for 89.9% of the whole hibernation period. That is, the whole hibernation period is composed of several torpor–arousal hibernation bouts [64]. For late torpor (LT), ground squirrels were weighed and sampled once they had hibernated for 2 months, entered a new hibernation bout and torpor continuously for at least 5 d, and their body temperature (Tb) was stabilized at 5–8 °C. For interbout arousal (IBA), when the ground squirrels had hibernated for 2 months, entered a new hibernation bout, were fully aroused, and their body temperatures (Tb) had returned to 34–37 °C, the sample was taken within 12 h. It should be noted that we typically provided food and water during the whole hibernation period (we dissected the gastrointestinal tract of Daurian ground squirrels and found that there were no fresh contents in the stomach and small intestine during torpor, only fermented, almost black, contents in the cecum, but the gastrointestinal tract was full of fresh food residue during the interbout arousal period), i.e., Daurian ground squirrels were in intermittent food and water deprivation during the whole hibernation period (no food and water during torpor, ate food and drank water during the interbout arousal period).

### 4.2. Tissue Sample Collection and Preparation

#### 4.2.1. Colonic Sample Collection and Preparation

For collection of intestinal sample, ground squirrels were given abdominal anesthesia via injection with 20% ethyl carbamate at a dose of 1 mL/100 g (after weighing). After 15–30 min, when experimental animals were fully anesthetized, they were placed on the anatomical table and a midabdominal incision was made for dissecting and separating of colonic tissues. A 3 cm length of colonic tissue was sequentially cut and a 0.5 cm length of tissue adjacent to the proximal colon was placed into a 2 mL cryopreservation tube with 2.5% glutaraldehyde, fixed at room temperature for 2 h, and then transferred to a refrigerator at 4 °C for preparing ultrathin sections for transmission electron microscopy. A second, 1 cm section of colonic tissue was placed into a 2 mL cryopreservation tube filled with 4% paraformaldehyde and stored in refrigerator at 4 °C. After overnight, the colonic tissue was removed from the 4% paraformaldehyde solution, and then embedded with melted paraffin wax into tissue wax blocks using specific embedding molds (JB-P5, Wuhan Junjie Electronics Co., Ltd, Wuhan, China). After this, the trimmed tissue wax blocks were placed in a paraffin sectioning machine (RM2016, Shanghai Leica Instrument Co., Ltd, Shanghai, China) and sliced in 5 μm thick sections after fixation, extraction, dehydration (Donatello, DIAPATH, Martinengo, Italy), transparency, and wax immersion. The slices were floated at Frozen platform (JB-L5, Wuhan Junjie Electronics Co., Ltd.) in warm water at 40 °C to flatten the tissue. Then, the slides were picked up and placed in the oven (GFL-230, Tianjin Laibo Rui Instrument Equipment Co., Ltd., Tianjin, China) at 60 °C to bake the slices. After baking, the slices were taken out and stored at room temperature for HE, AB–PAS, immunofluorescence, and immunohistochemistry staining. Finally, 1 cm segments of colonic tissue were inserted into 2 mL lyophilized tubes and placed in a −80 °C refrigerator for immunoblotting assay and ELISA kit assay.

#### 4.2.2. Serum Sample Collection

After the colonic tissue was collected, blood was taken immediately from the main abdominal vein. The blood was collected, allowed to stand for 30 min, then centrifuged at 4 °C and 3000 rpm for 15 min, after which the first supernatant was collected. Samples were then centrifuged again at 4 °C and 15,000 rpm for 5 min, and a second supernatant was collected. Supernatant samples were transferred into 200 μL centrifuge tubes containing 150 μL each, and then stored at −80 °C for inspection. After sampling, animals were euthanized by injection of an excess of 20% ethyl carbamate, a method that avoids unnecessary suffering [65].

### 4.3. Morphology of Colonic Mucosa

#### 4.3.1. HE (Hematoxylin–Eosin) Staining

Dewaxing: The slices (*n* = 3) were placed sequentially in xylene I (10023418, SCRC, Shanghai, China) for 20 min, xylene II for 20 min, 100% ethanol I (100092683, SCRC, Shanghai, China) for 5 min, 100% ethanol II for 5 min, and 75% ethanol for 5 min, in order, followed by rinsing with distilled water. Then, the sections were stained with hematoxylin solution (G1003, Servicebio, Wuhan, China) for 3–5 min, rinsed with distilled water, treated with hematoxylin differentiation solution, rinsed with distilled water, and then treated with hematoxylin and Scott’s tap water bluing reagent, and then a final rinse with distilled water. The slices were then sequentially dehydrated in 85% ethanol for 5 min, 95% ethanol for 5 min, and then stained with eosin dye (G1003, Servicebio, Wuhan, China) for 5 min.

Dehydration: The slices were placed into 100% ethanol I for 5 min, 100% ethanol II for 5 min, 100% ethanol III for 5 min, xylene I for 5 min, xylene II for 5 min, in order, and subsequently sealed with neutral gum (10004160, SCRC, Shanghai, China). Then, the slices were observed by microscope inspection (Nikon Eclipse E100, Nikon, Japan) and analyzed with Image-Pro Plus (IPP) 6.0 software (MediaCybernetics, Rockville, MD, USA). Subsequently, five visual fields were randomly selected from each slice, and the colonic mucosal thickness and crypt depth were measured using IPP software.

#### 4.3.2. AB–PAS (Alcian Blue/Periodic Acid-Schiff) Staining

The dewaxing procedure is the same as for HE staining. Once dewaxed, the sections (*n* = 3) were stained in AB–PAS staining solution C (G1049, Servicebio, Wuhan, China) for 15 min, rinsed with distilled water until colorless, then stained with AB–PAS B (G1049, Servicebio, Wuhan, China) for 15 min, and rinsed with distilled water twice. The sections were stained with AB–PAS A (G1049, Servicebio, Wuhan, China) for 30 min in the dark at room temperature, and rinsed with distilled water for 5 min. The dehydration procedure was also the same as for HE staining. Then, the slices were observed via microscope inspection, and images were analyzed with IPP 6.0 software. Then, five visual fields were randomly selected from each slice, and the number of goblet cells was measured using IPP software.

#### 4.3.3. Transmission Electron Microscopy

The samples (*n* = 3) were rinsed in 0.1 M phosphate buffer (pH 7.4) three times, 15 min each. **Post-fix**: Tissues were fixed with 1% OsO_4_ (Ted Pella Inc., Shanghai, China) in 0.1 M PB (pH 7.4) for 2 h at room temperature in the dark. After removing OsO_4_, the tissues were rinsed in 0.1 M PB (pH 7.4) three times for 15 min each. Then, the tissues were dehydrated at room temperature as followed: 30% ethanol for 20 min, 50% ethanol for 20 min, 70% ethanol for 20 min, 80% ethanol for 20 min, 95% ethanol for 20 min, and 100% ethanol for 20 min. Finally, the tissues were rinsed with 100% acetone (10000418, Sinaopharm Group Chemical Reagent Co., Ltd, Shanghai, China) twice, 15 min each time. **Resin penetration and embedding**: Treatments were as follows. Acetone: EMBed 812 (90529-77-4, SPI, Madison, WI, USA) = 1:1 for 2–4 h at 37 °C. Acetone: EMBed 812 = 1:2 overnight at 37 °C, pure EMBed 812 for 5–8 h at 37 °C. Then, pure EMBed 812 was poured into the embedding models and the tissues were inserted into the pure EMBed 812, and held at 37 °C overnight. **Polymerization**: The embedding models with resin and samples were moved into a 60 °C oven to polymerize for >48 h. Subsequently, the resin blocks were taken out from the embedding models for standby application. **Ultrathin section**: The resin blocks were cut into 60–80 nm thicknesses on an ultramicrotome (Leica UC7, Leica, Germany), and the tissues were fished out onto the 150 mesh cuprum grids with formvar film. **Staining**: samples were stained for 8 min with 2% uranium acetate saturated alcohol solution (avoiding light), and then rinsed in 70% ethanol three times, followed by rinsing in ultrapure water three times, staining with 2.6% lead citrate solution for 8 min (avoiding exposure to CO_2_), and again rinsing with ultrapure water three times. After drying on filter paper, the cuprum grids were placed into the grids board and dried overnight at room temperature. **Observation and image capture**: The cuprum grids were observed under transmission electron microscope (HT7800/HT7700, Hitachi, Japan), and images were taken. However, unavoidable biological errors may occur when using the ultramicrotome. For example, during the sectioning process, the ultramicrotome cuts the sections at a certain angle, which may lead to the destruction of some histological structures, resulting in some histological artifacts in the resultant maps. Therefore, we need to take this into account and acquire more evidence when analyzing the transmission electron microscopy results.

### 4.4. Evaluation of Colonic Mucosal Barrier Function and Regulatory Mechanism

#### 4.4.1. Immunofluorescence Analysis

The dewaxing procedure was the same as HE staining. **Antigen retrieval**: the slides (*n* = 3) were immersed in EDTA antigen retrieval buffer (pH 8.0) (G1206, Servicebio, Wuhan, China) and maintained at a sub-boiling temperature for 8 min, then stood for 8 min, followed by exposure to another sub-boiling temperature for 7 min. Slides were then washed three times with phosphate-buffered saline (PBS, pH 7.4) (G0002, Servicebio, Wuhan, China), followed by rocking for 5 min each. **Circled and Serum blocked**: obvious liquid was drained off and the objective tissue was marked with liquid blocker pen (G6100, Servicebio, Wuhan, China). Then, 3% BSA (G5001, Servicebio, Wuhan, China) was added to cover the marked tissue to block non-specific binding for 30 min. **Primary antibodies**: Slides were then incubated with one of several antibodies: occludin (1:100, Abcam, ab216327, Cambridge, UK), ZO-1 (1:50, Thermo Fisher Scientific, 2617300, Eugene, OR, USA), or E-cadherin (1:500, Abcam, ab231303, Cambridge, UK) (each diluted with PBS appropriately). Incubation was conducted overnight at 4 °C with placement in a wet box containing a small amount of water. **Secondary antibody**: Slides were washed three times with PBS (pH 7.4) in a rocker device (TSY-B, Servicebio, Wuhan, China) for 5 min each. Slides were then exposed to an appropriate secondary antibody: Cy3 conjugated goat anti-rabbit IgG (H + L) (1:300, Servicebio, GB21301, Wuhan, China), or Cy3 conjugated goat anti-rabbit IgG (H + L) (1:300, Servicebio, GB21303, Wuhan, China), at room temperature for 50 min in the dark. **DAPI counterstaining of nucleus**: The slides were washed three times with PBS (pH 7.4) on a rocker for 5 min each, followed by incubation with DAPI solution (G1012, Servicebio, Wuhan, China) (ready-to-use, no need to dilute) at room temperature for 10 min in the dark. **Spontaneous fluorescence quenching:** The slides were washed three times with PBS (pH 7.4) on a rocker for 5 min each with spontaneous fluorescence quenching reagent (G1221, Servicebio, Wuhan, China), and then washed in running tap water for 10 min. **Mount:** After pouring off the liquid slightly, the slides were covered with anti-fade mounting medium (G1401, Servicebio, Wuhan, China). Microscope detecting and images were collected by fluorescence microscopy. DAPI glows blue under UV at an excitation wavelength of 330–380 nm and emission wavelength 420 nm; CY3 glows red excitation wavelength 510–560 nm and emission wavelength 590 nm.

#### 4.4.2. Immunohistochemistry Analysis

The dewaxing procedure was the same as HE staining. **Blocking endogenous peroxidase activity**: the sections (*n* = 3) were placed in 3% hydrogen peroxide and incubated at room temperature in darkness for 25 min. Then the slides were placed in PBS (pH 7.4) and shaken on a decolorizing shaker three times for 5 min each. **Serum sealing**: 3% BSA was added to the circle to evenly cover the tissue, and then the tissues were sealed for 30 min at room temperature. **Primary antibody incubation**: the sealing solution was gently removed, and one of the primary antibodies was added, as follows, at the dilutions indicated: occludin (1:200, Abcam, ab216327, Cambridge, UK), E-cadherin (1:1000, Abcam, ab231303, Cambridge, UK), ZO-1 (1:50, Thermo Fisher Scientific, 2617300, Eugene, OR, USA), MUC2 (1:1000, Abcam, Cambridge, UK), claudin-1 (1:50, Thermo Fisher Scientific, 519000, Eugene, OR, USA), claudin-2 (1:100, Thermo Fisher Scientific, 516100, Eugene, OR, USA). Samples were prepared with PBS (pH 7.4), and certain proportions were added to the sections; after which, and the sections were placed flat in a wet box and incubated overnight at 4 °C. **Secondary antibody incubation**: the sections were placed in PBS (pH 7.4) and washed by shaking on the decolorizing shaker three times for 5 min each. After the sections were slightly shaken and dried, the tissues were covered with HRP-conjugated goat anti-mouse IgG (H + L)) (1:300, Servicebio, GB23303, Wuhan, China) or (HRP-conjugated goat anti-rabbit IgG (H + L)) (1:300, Servicebio, GB23301, Wuhan, China) secondary antibody from the corresponding species of primary antibody and incubated at room temperature for 50 min. **DAB chromogenic reaction**: The sections were placed in PBS (pH 7.4) and shaken on the decoloring shaker three times for 5 min each. Newly prepared DAB color developing solution (G1211, Servicebio, Wuhan, China) was added in the circle after the sections were slightly dried. The color development time was controlled under the microscope. The positive was brownish yellow. The sections were then rinsed with tap water to stop the reaction. **Nucleus counterstaining**: the sections were counterstained with hematoxylin stain solution (G1004, Servicebio, Wuhan, China) for about 3 min, washed with tap water, differentiated with hematoxylin differentiation solution (G1309, Servicebio, Wuhan, China) for several seconds, washed with tap water, treated with hematoxylin returning blue solution (G1340, Servicebio, Wuhan, China), and finally washed with running water. **Dehydration and mounting**: the sections were placed in 75% alcohol for 5 min, 85% alcohol for 5 min, absolute ethanol I for 5 min, anhydrous ethanol II for 5 min, n-butanol (100052190, China National Pharmaceutical Group Chemical Reagent Co., Ltd.) for 5 min, xylene I for 5 min, dehydrated and transparent, remove the sections from xylene and let them dry slightly, and sealed the sections with neutral gum. Microscope detecting and images were collected by fluorescence microscopy. Finally, we also performed a negative control experiment (negative control (all steps and conditions are the same except that the primary antibody is not added, and the secondary antibody is different)) as a way to improve the credibility of our results.

#### 4.4.3. Immunoblotting Analysis

The colonic tissue samples (*n* = 3) (0.1 g) were weighed and fully homogenized in RIPA buffer that contained protease and phosphatase inhibitors (Heart, WB053A, Xi’an, China). After 15 min of centrifugation at 4 °C and 15,000 rpm, the supernatants were removed and detected using a Pierce^TM^ BCA Protein Quantitation kit (Thermo Fisher Scientific, 23227, Waltham, MA, USA). The supernatants were then mixed with 1:1 *v*:*v* SDS loading buffer (100 mM Tris, 5% glycerol, 5% 2-β-mercaptoethanol, 4% SDS, and bromophenol blue, pH 6.8) in a 1:4 *v*/*v* ratio, followed by boiling, and then the protein supernatant was diluted to a final concentration of 2.5 µg/µL. Finally, the supernatants were stored at −20 °C for further analysis. Subsequently, protein samples of colon (25 μg) were loaded onto 10% SDS-PAGE gels, followed by electrophoresis at 80 V for 30 min and 120 V for 1 h. The proteins were electrically transferred to 0.2 μm pore polyvinylidene difluoride (PVDF) membranes (Millipore, IPVH00010, Merck kGaA, Darmstadt, Germany) at 20 V for 10 min, 40 V for 30 min, 60 V for 20 min. The membranes were blocked using 5% skim milk in TBST (containing 10 mM Tris-HCl, 150 mM NaCl, 0.05% Tween-20, pH 7.6) at room temperature for 1 h, followed by overnight incubation at 4 °C with a primary antibody: occludin (1:1000, Abcam, ab216327, Cambridge, UK), E-cadherin (1:1000, Abcam, ab231303, Cambridge, UK), FXR (1:1000, Abcam, abx131383, Cambridge, UK), SHP (1:1000, Abcam, ab32559, Cambridge, UK), Bcl (1:1000, Abcam, ab196495, Cambridge, UK) or Bax (1:1000, Abcam, ab32503, Cambridge, UK). The next day, the membranes were washed with 0.1% TBST for 3 × 10 min, followed by 1.5 h incubation at room temperature with horseradish peroxidase (HRP)-conjugated anti-rabbit (1:5000, Zhuangzhi, EK020, Xian, China) or anti-mouse (1:5000, Zhuangzhi, EK010, Xi’an, China) secondary antibodies, and the membranes were again washed with TBST (3 × 10 min). The resulting immunoblots were visualized using enhanced chemiluminescence reagents (Thermo Fisher Scientific, NCI5079, Waltham, MA, USA), according to manufacturer’s instructions. Blot quantification was conducted using version 1.8.0 of ImageJ software. The total protein staining of the gel was used as the standardization control for all blots.

#### 4.4.4. Enzyme-Linked Immunosorbent Assay (ELISA)

The kits (Fankew, Shanghai FANKEL Industrial Co., Ltd., Shanghai, China) were used to measure the permeability (*n* = 6) of the intestinal mucosal barrier according to manufacturer instructions. Indicators included the expression levels of TNF-α (F3056-B), IL-1β (F2923-B), IL-6 (F3066-A), IL-10 (F3071-B) and CRP (F2957-A) in serum and colonic tissues, respectively, the protein expression levels of claudin-1 (F40660-B), claudin-2 (F40655-B), ZO-1 (F40407-B), and MUC2 (F40647-B) in serum, and the expression levels of zonulin (F40391-B), DAO (F3678-A), I-FABP (F2927-B), and ET (F3243-B) in serum.

#### 4.4.5. Biochemical Testing

The expression level of D-lactate in serum was assessed using an ADS-F-T017 kit (Fankew, Shanghai FANKEL Industrial Co., Ltd., Shanghai, China). Reagents and samples (*n* = 6 6) were mixed and immediately reacted at 37 °C for 30 min in the dark, followed by transferring liquid to a 1 mL glass cuvette (optical diameter 1 cm), and the absorbance value was read at 450 nm. The concentration of D-lactic acid was calculated according to a standard curve obtained.

### 4.5. Statistical Analysis

Histomorphometric analysis of tissue sections was performed using Image-Pro Plus version 6.0 software, and immunoblot bands were quantified by ImageJ. All statistical tests were performed using SPSS statistical software version 20.0. Differences between groups were tested by one-way analysis of variance (ANOVA) and Fisher’s least significant difference (LSD). Finally, statistical analysis of all data was performed using GraphPad Prism version 8.0 software. Data for significance of differences tests between means were expressed as mean ± SEM (standard error of the mean). *p* < 0.05 was considered a statistically significant difference.

Research manuscripts reporting large datasets that are deposited in a publicly available database should specify where the data have been deposited and provide the relevant accession numbers. 

## 5. Conclusions

Hibernation altered the colonic mucosal morphology and the localization and expression of tight junction proteins and mucin in the colonic mucosa (compensatory response). Inflammation, and apoptosis–anti-apoptosis and FXR–SHP signaling may all be involved in counteracting the increased colon permeability that may occur during the torpor–arousal cycle. In addition, periodic interbout arousal during the long hibernation season of Daurian ground squirrels may play a role in inflammation correction.

## Figures and Tables

**Figure 1 ijms-23-09026-f001:**
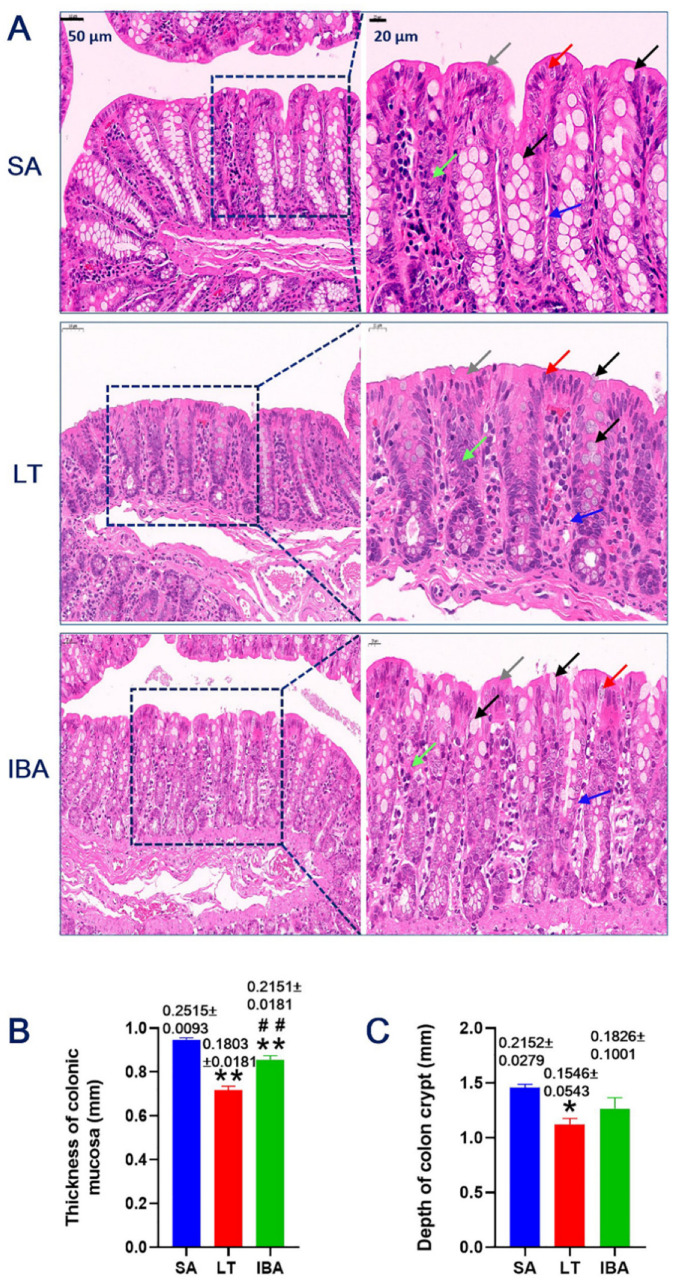
(**A**) HE (hematoxylin–eosin) staining of colonic mucosal tissue of Daurian ground squirrels by light microscope. Columnar epithelial cells (red arrows), goblet cells (GCs) (black arrows), basolateral membrane of subepithelial in surface (gray arrows), basolateral membrane of epithelia in crypt (blue arrows), lamina propria (green arrows). (**B**) Mucosal thickness measured by taking five random fields per section. (**C**) Depth of crypt measured by taking five random fields per section. SA, summer active; LT, late torpor; IBA, interbout arousal. Mean ± SEM. *n* = 3. * *p* < 0.05, ** *p* < 0.01, compared with SA group. ## *p* < 0.01, compared with LT group.

**Figure 2 ijms-23-09026-f002:**
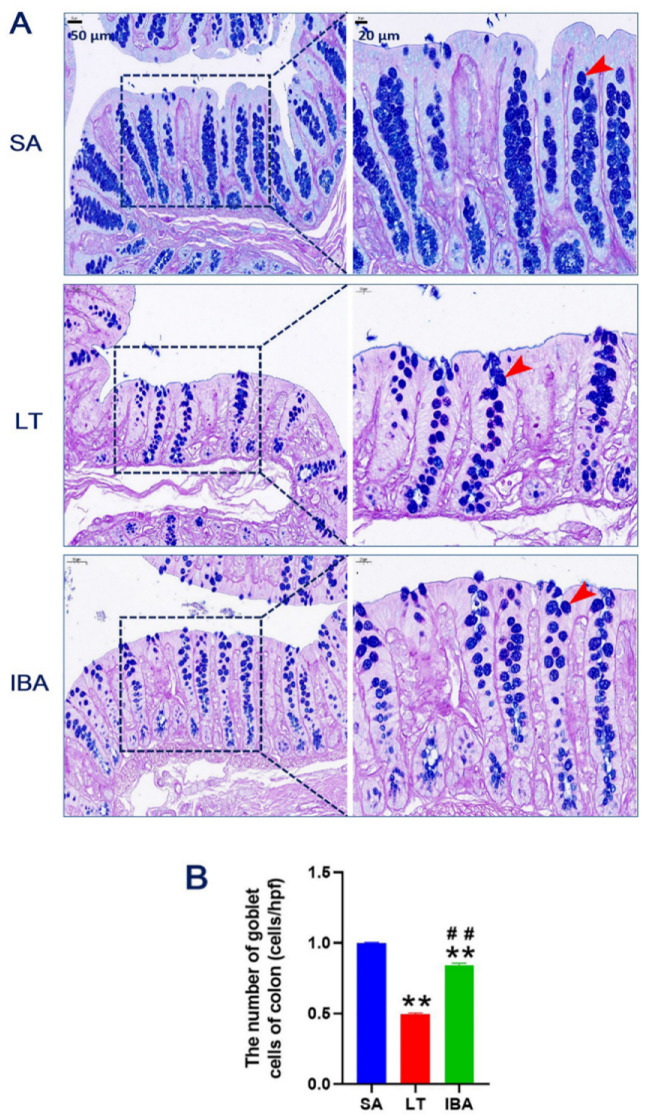
(**A**) AB–PAS (Alcian blue/periodic acid-Schiff) staining of colonic mucosal tissue of Daurian ground squirrels by light microscope. Goblet cells (GCs) (red arrows). (**B**) The number of GCs measured by taking five random fields per section. SA, summer active; LT, late torpor; IBA, interbout arousal; hpf, high power field (or high magnification field). Mean ± SEM. *n =* 3. ** *p* < 0.01, compared with SA group. ## *p* < 0.01, compared with LT group.

**Figure 3 ijms-23-09026-f003:**
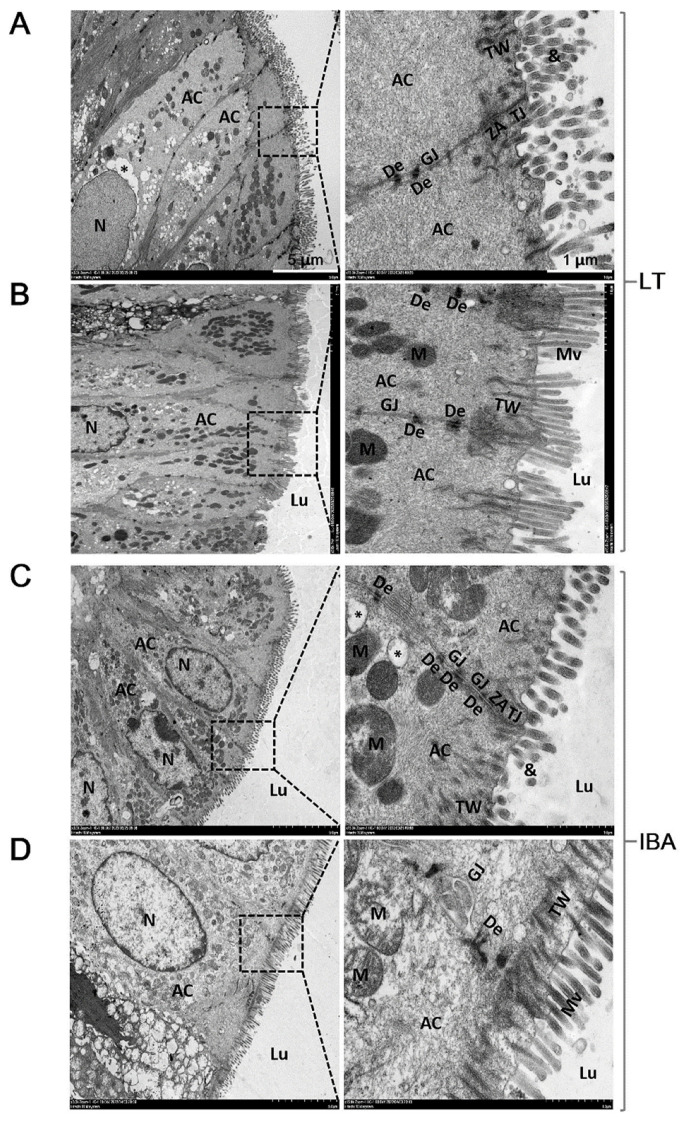
Transmission electron microscopy imaging of colonic mucosal epithelial cells in LT group (**A**,**B**) and IBA group (**C**,**D**) of Daurian ground squirrels. Colonic lumen (Lu), absorptive cells (AC), microvilli (Mv), microsomes (&), nucleus (N), mitochondria (M), terminal network (TW), tight junctions (TJ), zonula adherens (ZA), desmosomes (De), gap junctions (GJ). LT, late torpor; IBA, interbout arousal.

**Figure 4 ijms-23-09026-f004:**
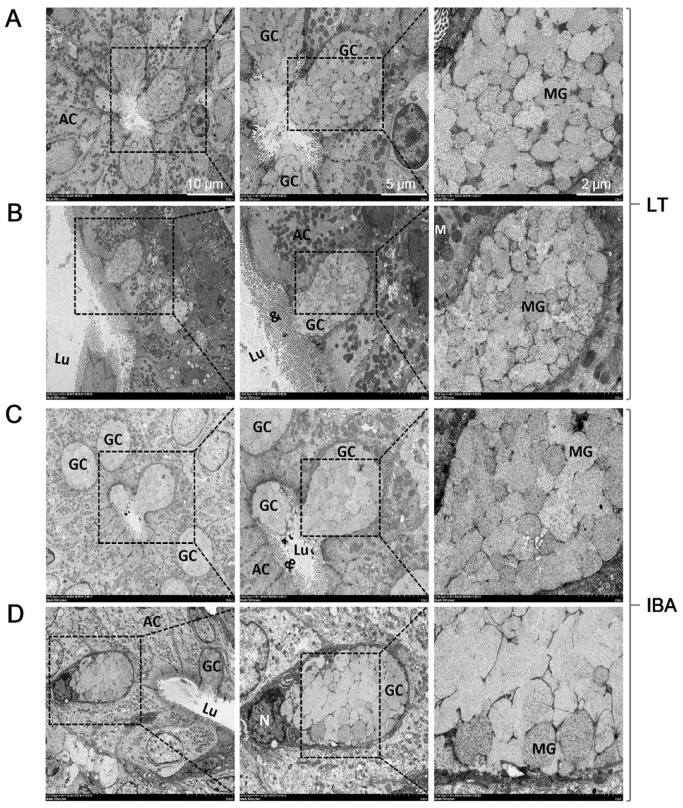
Transmission electron microscopy imaging of goblet cells in the colonic mucosa of the LT group (**A**,**B**) and the IBA group (**C**,**D**) of Daurian ground squirrels. Colonic lumen (Lu), goblet cells (GC), mucus granules (MG), microsomes (&), absorptive cells (AC), mitochondria (M), nucleus (N). LT, late torpor; IBA, interbout arousal.

**Figure 5 ijms-23-09026-f005:**
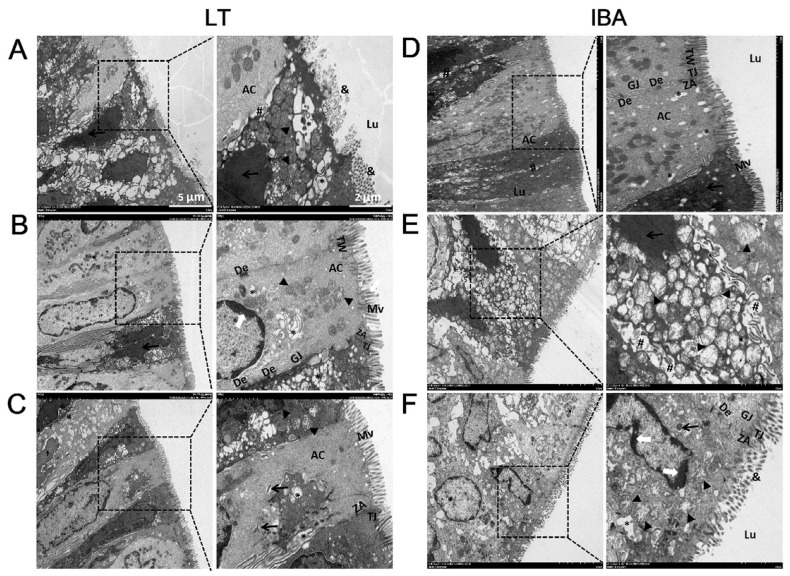
Transmission electron microscopy imaging of apoptotic cells in the colonic mucosa of the LT group (**A**–**C**) and IBA group (**D**–**F**) of Daurian ground squirrels. Colonic lumen (Lu), microvilli (Mv), microsomes (&), terminal network (TW), tight junctions (TJ), zonula adherens (ZA), desmosomes (De), gap junctions (GJ), enlarged cell–matrix gap (#), intranuclear chromatin agglutination, marginalization (white thick arrow), mitochondrial swelling, aggregation, or disruption of cristae structure (black triangle), vacuolated cellular debris in the nuclear periphery and cytoplasm (*), apoptotic bodies (black arrows). LT, late torpor; IBA, interbout arousal.

**Figure 6 ijms-23-09026-f006:**
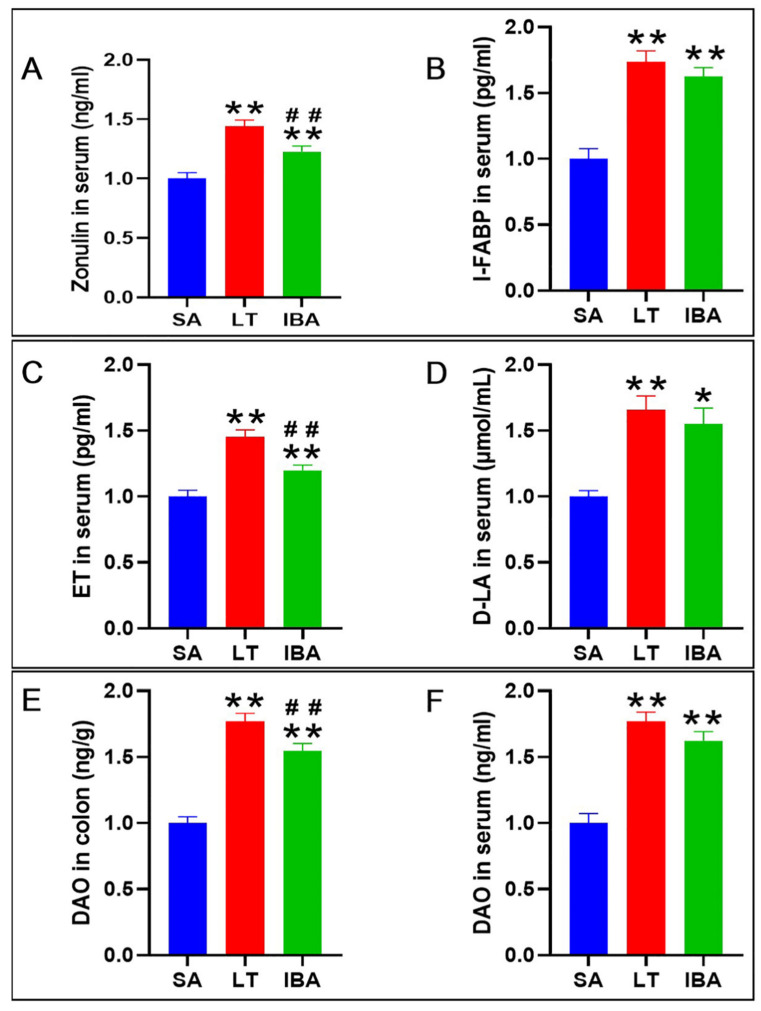
Concentrations of zonulin (**A**), I-FABP (**B**), ET (**C**), and D-LA (**D**) in serum and the activity of DAO in colonic tissues and serum (**E**,**F**) of Daurian ground squirrels. I-FABP (enteric fatty-acid-binding protein), ET (endotoxin), D-LA (D-lactic acid), DAO (diamine oxidase). SA, summer active; LT, late torpor; IBA, interbout arousal. Mean ± SEM. *n =* 6. Compared with SA group, * *p* < 0.05, ** *p* < 0.01. Compared with LT group, ## *p* < 0.01.

**Figure 7 ijms-23-09026-f007:**
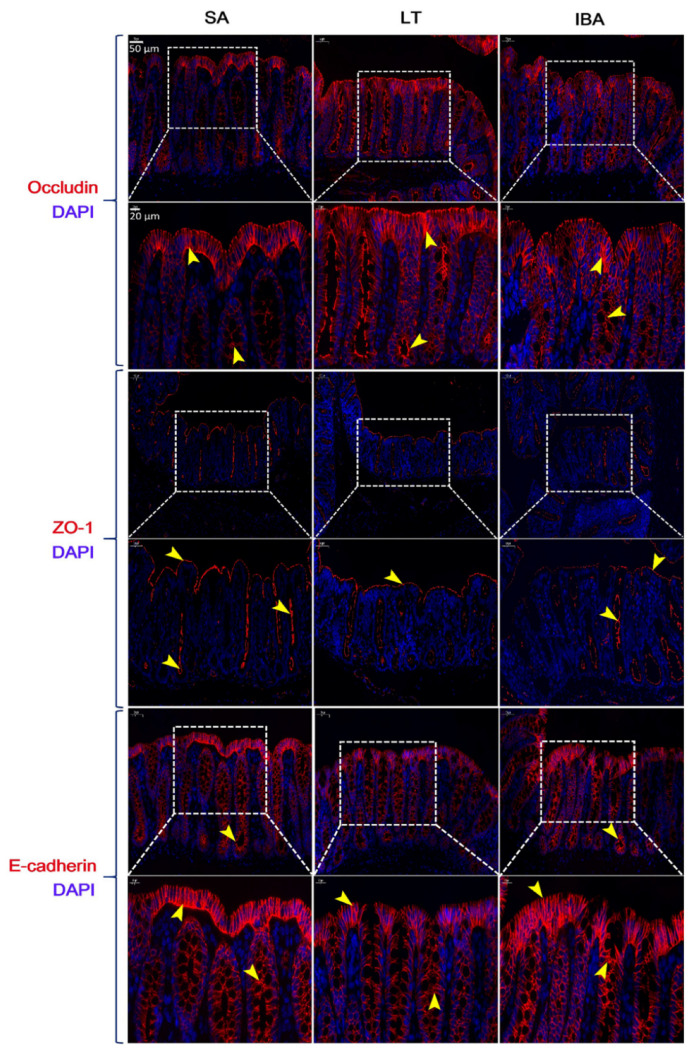
Distribution of occludin, E-cadherin, and ZO-1 proteins in colonic mucosa of Daurian ground squirrels. DAPI (blue) marks the nucleus, CY3 (red) marks occludin, E-cadherin, and ZO-1. Positive expression (yellow arrows). *n =* 3. SA, summer active; LT, late torpor; IBA, interbout arousal.

**Figure 8 ijms-23-09026-f008:**
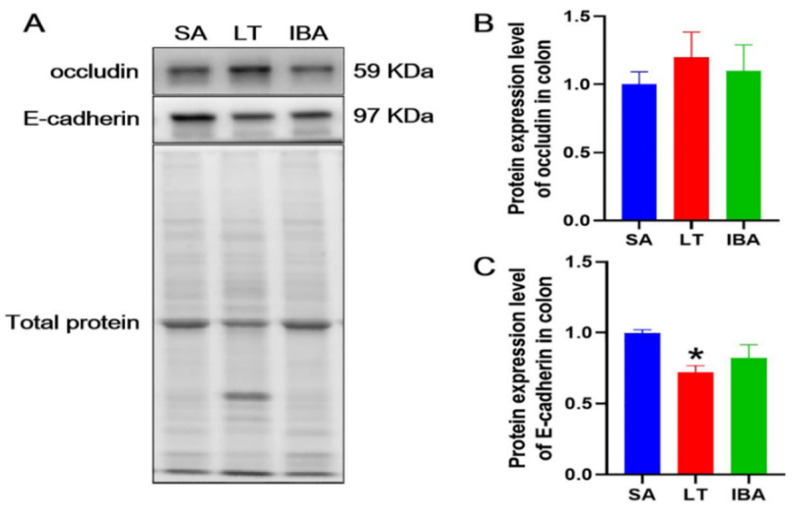
(**A**) Representative immunoblot images of occludin and E-cadherin in the colon of Daurian ground squirrels. (**B**) Statistical graph of protein expression level of occludin. (**C**) Statistical graph of protein expression level of E-cadherin. SA, summer active; LT, late torpor; IBA, interbout arousal. Mean ± SEM. *n =* 3. * *p* < 0.05, compared with SA group.

**Figure 9 ijms-23-09026-f009:**
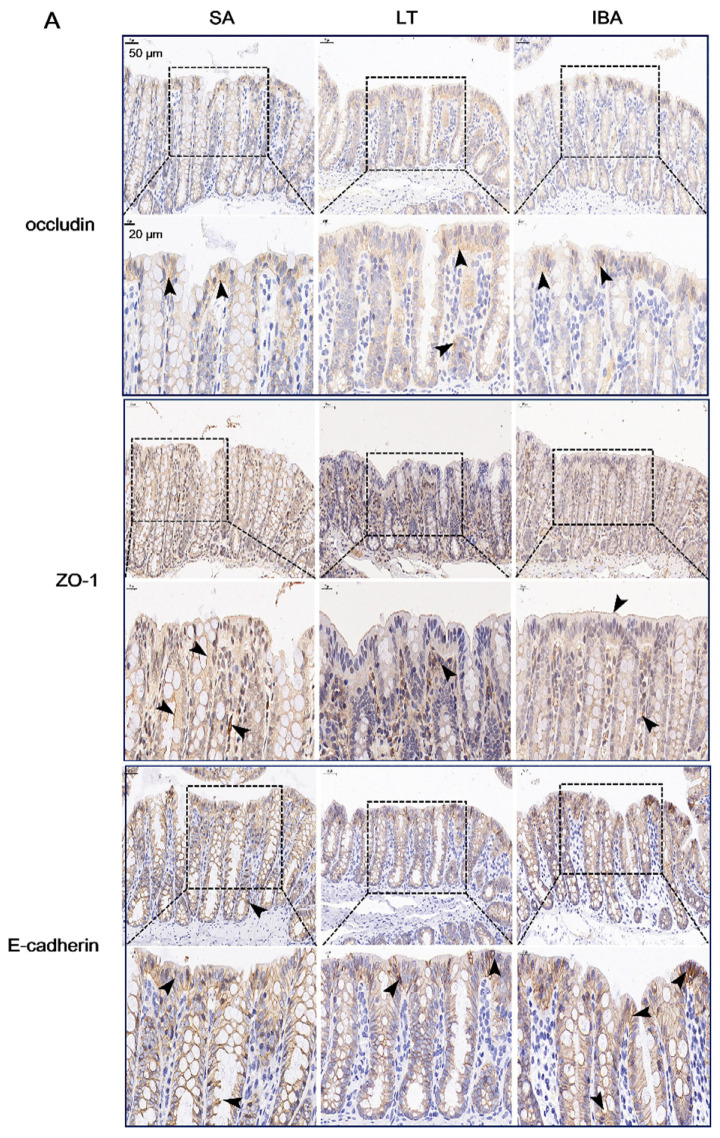
(**A**) Immunohistochemical staining of colonic mucosal occludin, ZO-1, and E-cadherin proteins in Daurian ground squirrels. (**B**) Immunohistochemical staining of colonic mucosal claudin-1, claudin-2, and MUC2 proteins in Daurian ground squirrels. (**C**) Negative control (all steps and conditions are the same, except that the primary antibody is not added and the secondary antibody is different). Hematoxylin-labeled the nucleus (blue), DAB staining positive when brownish yellow (black arrows). *n =* 3. SA, summer active; LT, late torpor; IBA, interbout arousal.

**Figure 10 ijms-23-09026-f010:**
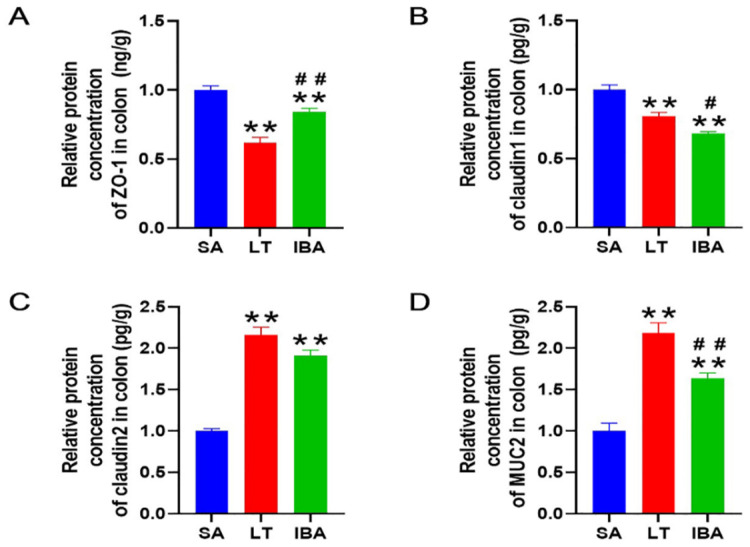
ELISA assay results for (**A**) protein concentrations of ZO-1, (**B**) protein concentrations of claudin-1, (**C**) protein concentrations of claudin-2, (**D**) protein concentrations of MUC2. SA, summer active; LT, late torpor; IBA, interbout arousal. Mean ± SEM. *n* = 6. Compared with SA group, ** *p* < 0.01. Compared with LT group, # *p* < 0.05, ## *p* < 0.01.

**Figure 11 ijms-23-09026-f011:**
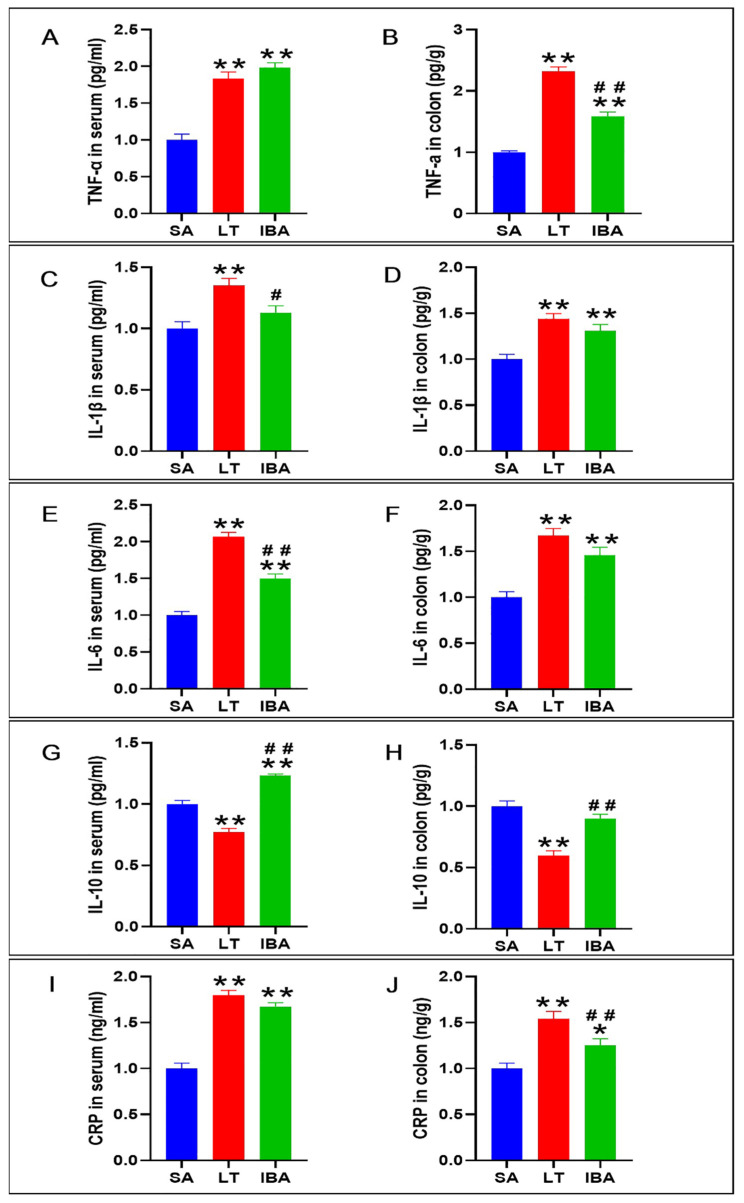
Detection of inflammatory factors in serum and colon of Daurian ground squirrels. (**A**) The concentrations of TNF-α (tumor necrosis factor-α) in serum. (**B**) The concentrations of TNF-α in colon tissue. (**C**) The concentrations of IL-1β (interleukin-1β) in serum. (**D**) The concentrations of IL-1β in colon tissue. (**E**) The concentrations of IL-6 (interleukin-6) in serum. (**F**) The concentrations of IL-6 in colon tissue. (**G**) The concentrations of IL-10 (interleukin-10) in serum. (**H**) The concentrations of IL-10 in colon tissue. (**I**) The concentrations of CRP (C-reactive protein) in serum. (**J**) The concentrations of CRP in colon tissue. SA, summer active; LT, late torpor; IBA, interbout arousal. Mean ± SEM. *n* = 6. Compared with SA group, * *p* < 0.05, ** *p* < 0.01. Compared with the LT group, # *p* < 0.05, ## *p* < 0.01.

**Figure 12 ijms-23-09026-f012:**
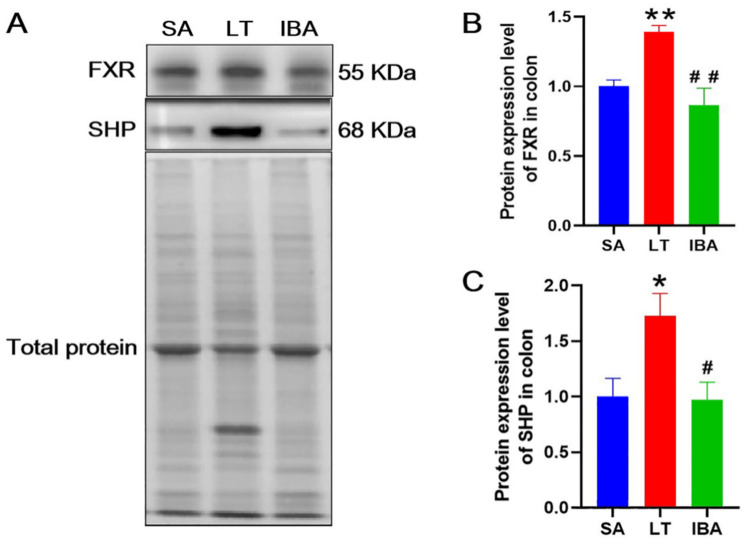
(**A**) Representative immunoblot images of FXR and SHP proteins in colon tissue of Daurian ground squirrels. (**B**) Statistical graph of protein expression level of FXR (farnesoid X receptor). (**C**) Statistical graph of protein expression level of SHP (small heterodimer partner). SA, summer active; LT, late torpor; IBA, interbout arousal. Mean ± SEM. *n* = 3. Compared with SA group, * *p* < 0.05, ** *p* < 0.01. Compared with LT group, # *p* < 0.05, ## *p* < 0.01.

**Figure 13 ijms-23-09026-f013:**
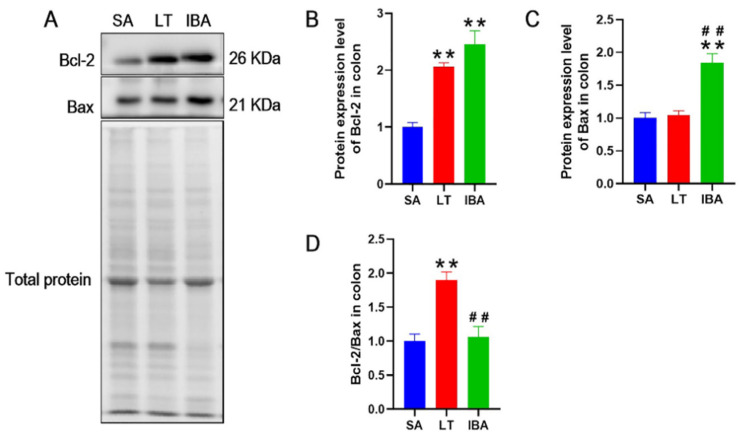
(**A**) Representative immunoblot images of Bcl-2 and Bax proteins in colon tissue of Daurian ground squirrels. (**B**) Statistical graph of protein expression level of Bcl-2. (**C**) Statistical graph of protein expression level of Bax. (**D**) Statistical graph of Bcl-2 /Bax ratio. SA summer active; LT, late torpor; IBA, interbout arousal. Mean ± SEM. *n* = 3. Compared with SA group, ** *p* < 0.01. Compared with LT group, ## *p* < 0.01.

## Data Availability

Not applicable.

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
