# Peer review of "Dynamic Changes in Colonic Structure and Protein Expression Suggest Regulatory Mechanisms of Colonic Barrier Function in Torpor–Arousal Cycles of the Daurian Ground Squirrel"

_ijms, 2022, doi:10.3390/ijms23169026_

Round 1

Reviewer 1 Report

The results presented in this study make a contribution to our current understanding of how hibernation in mammals, which is associated long periods of metabolic depression and for some species, fasting, affects structure and potential function of the gastrointestinal tract.  Previous studies have focused primarily on the small intestine and the first portion of the large intestine (cecum), but to date there is very little information on seasonal and metabolic state-specific changes distal to the cecum, that is, the distal colon which is the focus of this study.  Studies on the effects of hibernation on the gastrointestinal tract have also not been carried out previously with Daurian ground squirrel, which do not fast during hibernation – according to the authors, they were provided food and water in this study and they ate during interbout arousal periods (that needs to be made clear in the Methods).  So there is some new information presented in this manuscript.

However, there are some serious problems with the manuscript in its present form, and it requires extensive revisions to correct statements that are incorrect, fix multiple errors, carry out additional studies especially to increase sample sizes (which are mostly n=3) and improve the writing. 

I describe below two major concerns I have with the manuscript that should be addressed by the authors, and then list a number of other concerns, and suggestions, to improve the clarity and presentation of the data presented and the writing. 

Major comments 

1)Despite what is implied in your title and in various places throughout the manuscript there are no data presented that measure gut barrier function or specifically, colonic permeability - that is, there are no functional studies, rather you provide only histological measurements and changes in protein expression in colon tissue, and changes of some proteins and molecules and serum.  The only exception to this is at the very end of the Discussion where do you acknowledge the lack of functional permeability measurements, but at that point readers will have already read the preceding text (and title of the manuscript) which generally seems to imply that you obtained permeability measurements. Therefore, the title should be revised, because without functional measurements of barrier function/gut permeability, and especially interventional studies to demonstrate that one or more histological or protein expression changes influenced permeability, it is not possible to state that you demonstrated the regulatory mechanisms that influence the seasonal (summer vs. winter) and metabolic-state specific (IBA vs. torpor) permeability changes in that you presume happen in Daurian squirrels. You can certainly discuss your results as possible regulatory mechanisms that may influence the permeability changes shown in other studies, such as changes in protein expression that are known to regulate barrier function in other animal studies, but experiments in this manuscript have not revealed the mechanisms per se.  A suggestion for a title change is “Dynamic changes in colonic structure and protein expression suggest regulatory mechanisms of colonic barrier function in torpor-arousal cycles of the Daurian ground squirrel (Spermophilus dauricus)” (also note there is no need to use quotation marks around torpor-arousal). 

2) The second  major concern with this study is that most of the statistical analyses (especially the histological ones) are based on data from a sample size of 3 squirrels per group.  Of the 7 figures with data for which statistics are shown, only one Figure (11) indicates N=6, the rest are n=3.  This is a problem because especially for non-laboratory bred (that is, wild) animals, N=3 is too low to draw definitive conclusions due to interindividual variability, even if the statistical analysis suggested there is a “significant difference” at the P<0.05 level.  There is also no statement in Methods under Statistical Analysis that indicates power analyses were done that would suggest that n=3 is sufficient to represent most animals in these different states.  I therefore strongly suggest that if any additional fixed tissues are available, that you use them to analyze a few more samples, so that you have at least N=4-5 squirrels or greater for each group.  For example, I noticed that the legend of Fig. 6 indicates N=6 animals per group so perhaps there are histological samples from an additional 1-3 per group to add to data sets with only n=3.  Otherwise, with such a low sample size, readers may question the validity of some of the conclusions drawn (this is the case for me).

Other comments

The authors have done a reasonable job in reviewing what is currently known about this field from prior publications although there are some errors in the text that describes those studies (I noted many of them below).  However, there is one publication highly relevant to this manuscript that is not referred to and should be considered by the authors in the background information (Introduction) and certainly in the Discussion.  This will provide more context of previously known information on the effect of hibernation on gastrointestinal tract structure and protein expression, and the potential or measured functional effects that those changes may be indicative of:

Dill-McFarland et al. Hibernation alters the diversity and composition of mucosa-associated bacteria while enhancing antimicrobial defence in the gut of 13-lined ground squirrels. Molecular Ecology 23: 4658–4669, 2014.

Briefly, this article showed that compared with summer squirrels, protein expression of MUC2 in 13-lined ground squirrel cecal tissue was higher after 1 month of hibernation, and then expression fell in late winter hibernators so that it was similar to summer (> 4 months of hibernation) (Fig. 3 of Dill-McFarland et al.).  The hibernating squirrels had not eaten after entering the cold room in the fall (September-October) so these hibernating squirrels do not eat during IBAs.  Values in spring squirrels (after 2 weeks of refeeding) were similar to Early Winter.  Expression of this goblet cell protein that produces the mucus that is secreted into the apical space in the gut lumen and within the crypt lumen varies in 13-lined ground squirrel large intestine (cecum) from summer to winter and between torpor and IBA.  The Dill-McFarland paper et al. also demonstrated (see Table 3) seasonal and state-specific changes in crypt dimensions, the density of goblet cells (GC) per crypt, as well as cecal epithelial cell protein expression of TLR4 and TLR5, two receptors whose functions are associated with epithelial inflammation/protection in other mammals. 

Line 226: Consider using “colonocytes” instead of the term “enterocytes” when referring to colonic epithelial absorptive cells.  GI physiologists generally use “enterocytes” when referring to epithelial cells in the small intestine.

In the first portion of Methods under Animals and Groups, it should be explicitlyy stated whether the Daurian squirrels had food and water available to them throughout the hibernation season- presumably they did not, as Introduction says that these squirrels “eat and drink again during interbout arousal”.  If food was available, did you record food intake rates during IBA periods?  Were there body mass differences between the torpor and IBA groups studied, or the duration of euthermic periods during IBA?  These pieces of information are important because changes in colonic epithelial structure likely occurs during the IBAs (and not during the metabolically depressed state of torpor).  If some squirrels were above ~ 34 C for 12 hours or less, whereas others were at high body temperature in an IBA for closer to 24 hours, this could potentially affect the expression of epithelial cell proteins and/or mucosal or crypt dimensions as well as goblet cell mucus accumulation, and should be taken into account when you discuss your results.

Methods section should specify what is meant by “late torpor” this study; line 715 indicates tissues were collected when squirrels were in continuous torpor “for at least 5 d”, but without information on torpor bout lengths after two months of hibernation it is not clear what point in the hibernation cycle that is.  What is the average torpor bout lengths for Daurian squirrels after two months of hibernation?

Page 2, lines 92-94:  the statement that “mucus secreted by the colonic epithelial goblet cells was reduced during hibernation in Arctic ground squirrels” from reference [30] is misleading and should be modified.  First, there was no direct measurement of amount of mucus secreted from colonic goblet cells, only histological observations suggesting the that amount of mucus in epithelial goblet cells (i.e., GC size) was lower in squirrels in hibernation (presumably this means “in torpor” in ref 30) from that in the warm (presumably referring to interbout arousal hibernators) or active (summer) Arctic ground squirrels.  This observation could indicate that mucus was produced and accumulated in goblet cells during interbout arousal and then was secreted during that time (which seems a logical response to rewarming when metabolism rises again), such that when squirrels re-entered torpor, the goblet cells had less mucus contained in the cells as it was secreted during the prior IBA period. Because the histological data presented in that study provides no information on secretion itself nor the dynamics of production/secretion during torpor-arousal cycles, I suggest this sentence be modified to reflect only what the author states in ref. 30, such as “…the amount of mucus observed in large intestine goblet cells was lower when hibernating squirrels were in torpor from the amount observed in the warm hibernators and in active squirrels”.  

Methods: There is no information on whether food was present in hibernating squirrel cages, nor whether any food was eaten; if food was eaten, an indication of amount of food is would be very helpful to know.  Introduction states Daurian ground squirrels eat and drink during hibernation which is different from most other fat storing hibernators. 

Line 93: With regard to information in the Introduction regarding prior hibernation studies that examine intestinal permeability, I suggest you revise the sentences that start on line 93 and end on line 9 to correct several errors and cite other references that appear in your bibliography.  One error is that your citation to ref. 32 was a study not carried out with hibernating animals, rather it was cultured epithelial cells, so that citation should be deleted as it is not relevant. 

Lines 92-97: Please also be aware that use of FITC-dextran as a permeability marker cannot distinguish which part of the GI tract has altered permeability, as it is given orally and could therefore affect permeability in several parts of the gut (so you should revised the text to remove the implication that the FITC-dextran results are restricted to the small intestine or cecum specifically). 

Line 96:  there appears to be a word missing after “immunohistochemical”.  Also, please note that in addition to the study in reference 31 that directly examined gut macromolecular permeability in summer and hibernating squirrels using FITC-dextran (not FITC-glucoside) there are other earlier studies using the Ussing chamber technique that also show that permeability in the hibernator small intestine (torpid and IBA) squirrels is higher compared to summer squirrels as indicated by transepithelial conductance, which is a measure of total passive ion flow from serosal to mucosal sides of the intestinal epithelium.  Two of those studies are cited in your bibliography so you could incorporate these in this section as well.  In total, I recommend you consider modifying lines 93-99 with the appropriate citations and corrections to read:

“Evidence from 13-lined ground squirrels suggests there is higher gut permeability during the hibernation season compared to summer squirrels.  Transepithelial ion conductance measurements suggest that passive ion flow across the small intestine is elevated during hibernation [29, 54] and the luminal to blood movement of the macromolecular marker FITC-dextran is also elevated, which also indicates total gut permeability increases in hibernation [31].  Expression of the tight junction protein occludin in ground squirrel cecum including its phosphorylated form is greater in hibernating than in summer squirrels, as is localization of occludin in the apical intercellular spaces where it can regulate tight junction permeability [31].”

With regard to your sentence on lines 97-99, please note that at least in the 13-lined ground squirrels there is no direct evidence that increased permeability during hibernation is “adaptive”.  In fact, increased gut permeability during fasting in many animals can be a maladaptive response (see your citation 95) to the chronic absence of food in the gut lumen, and although it occurs in long-term fasted hibernators there may be other compensatory changes (such as the occludin results) that counteract that possibly deleterious change.  I would therefore suggest you consider altering that sentence to something like:

“Hence, from these studies it is evident that there are a series of changes that occur in the  gastrointestinal tract of hibernating mammals that fast during winter, some of which may be deleterious (e.g., increased permeability [95]) and others that may be compensatory adaptations to the permeability change (e.g., occluding expression and localization).” 

Acronyms should be defined the first time they are used in a manuscript.  In  your manuscript, many acronyms are mentioned such as those on p. 8 beginning on line 253 with no indication of what they refer to.  Please include the full name of a word with its acronym in parenthesis at the first appearance in the manuscript.

Fig 1 data and associated text:

Line 143: when looking at Fig 1, I cannot see the evidence for “goblet cells in LT and IBA groups tended to transfer to the upper part of the crypt or even the surface epithelium compared to SA”.  I see goblet cells on the surface in SA and IBA but very few in that location in LT.  As mentioned in Major Comments, I am concerned that a sample size of n=3 animals is not enough be make such a conclusion for all Daurian squirrels?  I suggest removing this statement, especially since you referred to it as a tendency.

Line 157 and elsewhere in the manuscript:  you use the term “basement membrane” of epithelial cells but I think you mean “basolateral membrane”.  I have never heard of “basement membrane” being used to describe the non-apical membrane of a colonic epithelial cell. 

Fig 2 data and associated text:

It is not apparent to me from the representative sections shown in the figure that number of GC cells per field are substantially lower in LT than SA (or IBA).  In fact the numbers of GC in LT per crypt look even greater than IBA.  What does the statistical analysis show when you normalize the number of GC to crypt length?   I strongly suggest you do that analysis, and consider that the basis of normalization may make a difference in your conclusions rather than your current method of normalization reflecting a functionally significant change in the animals colons.  Also, do you think the duration of the IBA period is enough for there to be that much change in GC proliferation in the stem cell zone and migration up and down within the crypt itself?  You might try to find data from other rodents on how long it takes for a newly generated GC to reach the surface epithelial layer or migrate deeper to the base of the crypt.  Data are available for 13-lined ground squirrel small intestine in this article” Carey and Martin, 1996 Am J Physiol 271:G804-13,1996. Preservation of intestinal gene expression during hibernation DOI: 10.1152/ajpgi.1996.271.5.G805.

Fig 3 data and associated text:

Lines 188-199: “some epithelia were covered with microvilli that were structurally disrupted, shortened in length, or even disintegrated into microsomes (Figure 3A, 3C)”.  The micrograph images of the disrupted microvilli in Fig. 3 look to me as evidence for uneven sectioning, that is, an artifact of the sectioning process that varied for some sections, but not others, which cut the microvilli at angles losing the full structure of the microvillus.  How did you rule out a sectioning artifact that could lead to your conclusion of a biological reason for the images?  There is no information in Methods that shows you considered this (common) histological artifact to explain the results.

Line 203: here and elsewhere in other figures the term “tubular lumen” is used for the colonic lumen.  In my experience “tubular lumen” is not used in intestinal physiology, rather it is the common term for the lumen of kidney tubules.  I suggest change to “colonic lumen”. 

Fig 4 data and associated text:

Line 213:  I’m not certain what the phrase “almost squeezed” means with regard to organelles in LT and IBA groups.  Is that a cell biological term? I’ve not heard of it before, and it is difficult to understand the meaning from Fig. 4.

Fig 5 data and associated text:

Line 227-229: statements are made about apoptotic cells in LT and SA exhibiting abnormal morphology that “intuitively” (I’m not sure what is meant by “intuitively here) showed that they were apoptotic cells.  However, there are no images from SA animals for comparison, and no data presented on proportions of apoptotic cells in each animal group.  Therefore the conclusion made here is questionable.  Are you saying that SA squirrels never show colonocyte apoptosis, but it does occur during hibernation?

Also, here and elsewhere please consider rearranging all panels so that the lumen faces the same direction.  It is very distracting to try to compare images when the apical surfaces (lumen) are oriented both left, right, up and down in the same panel.

Fig 6 data and text:

Legend, line 265: change “Contents” to “Concentrations”

Consider switching panels D and F in so that the DAO graphs are adjacent to each other.

Fig 7 data and text:

Line 277: localization of “occludin and E-cadherin the colonic mucosal surface… in the apical area of colonic mucosal epithelial cells…”.  These are tight junction proteins that localize to the basolateral (or “intercellular” membranes in the upper part of tight junctions between adjacent epithelial cells - close to the apical surface but not on the apical membranes of the cells themselves.  Please modify this wording to reflect this distinction, or if you have published sources indicating they are these proteins can also be found on the apical membranes of colonic epithelial cells, it would be helpful to include a relevant citation.

I found most of the immunostaining micrographs shown in Figs 9A and 9B to be quite poor in quality, with abundant distribution of brown staining throughout many parts of the cells and crypts.  This may very likely be due to nonspecific staining either from the secondary antibodies or DAB.  How did you control for nonspecific staining to ensure there was no brownish-yellow in the absence of primary antibodies?   I could find no text related to confirmation of positive reactions by using appropriate negative controls for immunostaining in the “Immunohistochemistry” section of Methods.

Lines 299-301: “claudin2 and MUC2 in the colonic mucosa of the SA group were strongly localized in the upper part of the crypt, but both had migrated to the lower 300 part in LT and IBA groups”.  This is unclear, as cellular proteins do not migrate to other parts of the crypt structure.  Rather, after proliferation in the stem cell zone entire cells move in the apical or serosal orientation in the crypt.

As mentioned above, for data in figures 12A and B, it is very difficult to draw reliable conclusions regarding the role of FXR, SHP, Bax and Bcl-2 in colonic tissues based on n=3 animals per group.

Discussion

Line 416:  what do you mean by “…two important causes of intestinal permeability”?  Intestinal permeability is not a disease or dysfunction, rather simply a measurement of the degree of movement of molecules across the epithelial layer.  Do you mean to say “important causes of elevated intestinal permeability”?  Also, you cite ref 51 in this sentence but I could find nothing in the cited article about intestinal barrier dysfunction and impaired intestinal epithelial integrity being two causes of intestinal permeability, in fact the article does not contain the words “barrier dysfunction” or “integrity”.  I am also curious how you distinguish the difference between “barrier dysfunction” and “impaired intestinal epithelial integrity” – are they two distinct conditions?  From my understanding they are very close in meaning and are sometimes used interchangeably.

Lines 418-420: “Maintenance of the barrier function could be particularly important for hibernating animals because mammals lack nutrition after winter fasting, which can increase the permeability of intestinal epithelial cells”.  The logic of this sentence is not clear when you are studying a hibernator that eats during interbout arousals:  if the squirrels you used in the experiments ate food during IBAs, wouldn’t you predict that they should not be at risk for elevated gut permeability during hibernation?

Line 420:  The citation here makes no sense, ref. 52 deals with the lung, not the intestine.  It should be deleted and use instead an article that involved the intestine.

Lines 422-423: “we investigated the effects of hibernation on mucosal barrier…function (intestinal mucosal permeability),”.  This is incorrect, your study did not include measurements of mucosal permeability.

Lines 423-424: “we investigated the effects of hibernation…and mechanism (inflammatory factors, tight junction-related proteins, FXR, SHP and apoptosis-related proteins).”  This wording should be modified as you did not carry out interventional experiments to identify mechanisms, rather you measured changes in localization and abundance of a variety of proteins that could be related to gut  permeability, but the changes you observed could also be due to other aspects of colonic function.  If you had permeability measurements in your animals, and showed that it does change from summer to hibernation and from LT to IBA, then you see if any changes in these proteins correlated with changes in permeability, but even then you cannot say you have identified the mechanisms involved.  Also, you don’t really know yet if hibernation alters gut permeability in Daurian squirrels.

I suggest you modify that text to “….and changes in inflammatory factors, tight junction-related proteins, FXR, SHP and apoptosis-related proteins that could play a role in permeability changes that may occur during hibernation in Daurian ground squirrels.” 

Line 425, please modify this sentence, as “barrier function of the colonic mucosa” was not examined. 

Lines 426-430 show writing that is appropriate for what was examined in the study, although just because a protein level is greater or lesser than summer does not mean it is involved in a functional response such as barrier function.  This is particularly important because your sample size was exceedingly low (n=3) for most of the analyses.

Line 449-451:  “It has also been shown that complete atrophy of the jejunal  mucosa also occurs during hibernation in thirteen-line ground squirrels (Spermophilus tridecemlineatus) [54].”  This is not entirely correct; as shown in the reference you cite, the villus-crypt structure of the jejunum is still present during hibernation in this species, so atrophy is not “complete”, rather the amount of tissue is reduced (i.e., villi are shorter and crypt length is reduced) compared with squirrels that are feeding in summer.  I suggest modify to “…shown that the jejunal mucosa undergoes atrophy during hibernation in…”  Essentially, remove the word “complete”.

Also please note that the genus name of this squirrrel was changed several years ago to Ictidomys so the full scientific name is (Ictidomys tridecemlineatus).

Lines 457-464: You can also cite Dill-McFarland et al 2014 in this section; Table 3 in that studied shows that crypt length (refers to the same thing as the word “depth”) was reduced in early (1 month of hibernation) and in especially late winter (>4 months hibernation) compared with summer squirrels.  Because cecum is part of the large intestine as is the distal colon (which you studied), this may be of particular value to this part of the Discussion.

Line 456: “hibernation raises the relative abundance of Bacteroidetes and Verrucomicrobia in the cecum contents of thirteen-lined ground squirrels, while decreasing the relative abundance of Firmicutes, and these phyla contain species that can survive on host-derived substrates (e.g. mucins) [55].”  You should be more specific here in what species you are referring to, I suggest change to “…ground squirrels, while decreasing the relative abundance of Firmicutes.  Bacteroidetes and Verrucomicrobia contain species that can survive on host-derived substrates (e.g. mucins) [55].” 

Line 57-58:   “In addition, the depth of crypt in the colonic mucosa of ground squirrels was also significantly reduced”  I suggest you insert “Daurian” in front of ground squirrel here so readers are clear on which ground squirrel species you are referring to.

Lines 462-463:  “adaptive reduction of intestinal absorption and secretion functions, as a result of winter fasting.”  It is not clear why  you are calling this “adaptive” reduction, as there is no evidence to conclude it is indeed adaptive; I suggest removing the word “adaptive”.

Lines 474-478:  “prolonged fasting during hibernation may not affect the rapid recovery of the intestinal mucosa during the interbout arousal period, i.e., the intestinal mucosa seems to be well adapted to intermittent fasting, and this may be due to the rapid renewal rate of the intestinal epithelium during hibernation (3-5 days)[56, 57], and it is this rapid renewal that allows the intestine to respond quickly to various external factors during hibernation.”

In the above text, I think you are mixing the two different forms of hibernation responses which may be confusing to readers.  That is, some hibernators do not eat during hibernation (13-lined squirrels and I believe maybe bats (?), whereas Daurian squirrels do) and this difference in feeding may be responsible for some of the differences that various investigators (including yourselves) have found in the response of the GI tract to hibernation.  A value of your study is that you are working with hibernators that do eat during IBAs so you should emphasize that feature of your study.  Also there is an error in this text as the renewal rate of 3-5 days in 13-lined squirrels is for summer, not hibernating squirrels; for that you should cite this reference:  Carey HV and Martin SL. Am J Physiol 271:G804-13,1996. Preservation of intestinal gene expression during hibernation DOI: 10.1152/ajpgi.1996.271.5.G805

I suggest modifying the text in the following way to correct the errors and better explain the points you are making:

“For those species that do not eat during the hibernation season, prolonged fasting during hibernation may not favor the rapid recovery of the intestinal mucosa during the interbout arousal period.  In contrast, as displayed by Daurian ground squirrels, the intestinal mucosa may be well adapted to intermittent fasting, and this may be due to the resumption of cell renewal of the intestinal epithelium during IBAs (3-5 days) that has been demonstrated in other hibernators [Carey and Martin, 1996].”

Line 478: “Secondly, the results of AB-PAS staining showed that goblet cells in the colonic mucosa of ground squirrels in the SA, LT and IBA groups were regularly dispersed on the surface of the colonic mucosa and show high electron density (Figure 3A).”

This sentence does not seem to be correct. First you should refer to Fig. 2, not 3A.  Fig 2 shows that GCs are found not only on the surface epithelium of the colonic mucosa but throughout the crypt regions as well.  Please modify to reflect what the micrographs demonstrate.  The following sentences on lines 486-488 regarding the potential role of low metabolic rate and apoptosis in GC changes also do not make sense to me, and in any event are highly speculative so I suggest removing them.

Line 491: citations to the references [58-62] are not relevant as they have nothing to do with hibernation.

Lines 492-494:  here is a good place to cite the other study that examined MUC2 in hibernating mammals, that is Dill-McFarland et al. as mentioned at beginning of this review.

Line 501: Citing reference #69 for the statement that “Physiological parameters (e.g., temperature) are reduced during hibernation [69] is very strange, as that study is very specifically related to lung scaffolding; I suggest you choose one or more comprehensive hibernation references as references 1 and 3 in your bibliography.

Lines 503-504: “This may be the main reason why the number of goblet cells decreases during hibernation, but their secretion of MUC2 increases instead”.  This sentence implies you have a measure of changes in mucin secretion in the animal tissues, such as changes in thickness of the mucus lining, yet there are no data showing actual secretion of MUC2 (only immunohistochemistry and immunoblot).  Suggest change to ““This may explain why the number of goblet cells decreases during hibernation, but their expression of MUC2 increases”. 

Line 509: citation 71 is a study in insect midgut gut epithelium so not very relevant here; also the study notes that the cells examined “are responsible for all processes associated with digestion, secretion, and absorption…” (so more than simply absorption).  In mammals, to my knowledge it has not been shown that absorptive epithelial cells play a crucial role development of the physical barrier that maintains intestinal homeostasis more than other cells such as those in the crypts (which are the secretory cells).  Suggest modifying this text with a citation from a vertebrate or mammalian system that supports the claim that absorptive cells play that crucial role, or delete this sentence.

Line 522-523: “(1) The intact tight junctions in the colonic mucosa of ground squirrels during hibernation may be compensating for the adaptive changes in the morphology of the colonic mucosa.”   Why would tight junctions compensate for “adaptive” changes in the colonic mucosa?  Did you mean to say “contribute to the adaptive changes…? And as mentioned above, there need to be more direct evidence to refer to something as adaptive.  

Lines 525-526 and the following text: as mentioned earlier, you have not ruled out histological sectioning artifacts for the abnormal appearance of microvilli in hibernators in Fig 3, nor added any other evidence or other information suggesting this is a functional process that benefits the animals, so you cannot assign a functional role for the appearance of the tissue in the micrographs in those figures.  I suggest deleting this part of the Discussion (which is already very long).

Lines 555-557: “thirteen-lined ground squirrels [29], which undergo multiple hibernation periods interspersed with brief periods of awakening and feeding.” This statement is incorrect.  This species does not feed during interbout arousals.

Lines 569-571: “For hibernating animals, maintaining barrier function is particularly important because nutritional deficiencies in mammals increases the permeability of the intestinal epithelium, making the body more permeable to macromolecules during hibernation [95].” Since Daurian squirrels feed during interbout arousals, are you saying they still have nutritional deficiencies?  Or would they not have increased gut permeabilities because they do eat during hibernation, in contrast to other hibernators like 13-lined squirrels that fast during winter?

Lines 584-586: “Studies found increased immunohistochemical 584 staining for occludin in the small intestine and cecum of hibernating thirteen-lined ground squirrels, with particularly strong staining in apical cell spaces, and this result was supported by immunoblotting for hyperphosphorylated occludin in the cecum contents [31].”  This statement is not entirely correct.  Occludin is not on the apical membrane itself in these squirrels, rather it was found that “occludin staining was particularly strong in hibernators in the apical intercellular spaces, where tight junction protein complexes are located”.  You need to change “apical cell spaces to “apical intercellular spaces”, which specifies that occludin is in the tight junctions between adjacent cells, not on the apical membrane itself. 

Lines 597-598:  Keep in mind that MUC2 is not part of the tight junctional complex between epithelial cells that regulates epithelial permeability; it is a protein found in goblet cells.  MUC2 should be removed from this sentence.  This comment also applies to text in Lines 609-613; MUC2 should be removed from those sentences.

Line 625: change “awakening from hibernation” to “arousal from torpor during IBAs”.

Lines 625-626: “However, some studies have also found that the IL-10 levels decreased by almost half during hibernation and remain low during torpor and arousal.” What is the citation for this statement?  That is not correct for 13-lined squirrels (ref 33).  Please indicate what study you are referring to or deleted that sentence.

Lines 650-653:  I am not aware that FXR and SHP have been implicated in ischemia-reperfusion injury in the colon; this injury is well known to occur in the small intestine, but if you have references to studies that examined involvement of FXR and SHP influence the response to I/R injury in the colon, please including those citations.  This is the case for other instances in the Discussion or Introduction where you refer to protein changes that might be a response to ischemia-reperfusion, and if would be helpful to readers to know that this problem can occur in the distal colon as well.

The Discussion is exceedingly long, and in many places simply repeats the information already presented in Results.  Please use selective editing to reduce length of the Discussion (some suggestions for deletion are listed above); the Discussion currently is about 5.5 pages long, and a reduction to 4.0-4.5 would be beneficial.

Methods

In the Statistical Analysis section is the following text:

“Research manuscripts reporting large datasets that are deposited in a publicly available database should specify where the data have been deposited and provide the relevant accession numbers. If the accession numbers have not yet been obtained at the time of 935 submission, please state that they will be provided during review. They must be provided prior to publication. Interventionary studies involving animals or humans, and other studies that require ethical approval, must list the authority that provided approval and the corresponding ethical approval code.”

Clearly one of the authors must have copied this text directly from the journal’s Instructions for Authors.  It should be deleted and replaced by the relevant information that the Instructions request.

Reviewer 2 Report

Reviewer

In this study, the authors discribe the expression of tight junctions in the large intestine, ultrastructural investigation of epithelial cells, expression of pro-inflammatory factors, and expression of anti-apoptotic factors in the large intestine barrier in the torpor-arousal "cycle during hibernation using Daurian ground squirrels. This is a very interesting report because most of the previous reports have observed the expression of these factors in the small intestine, and have confirmed the expression of each factor in the large intestine in the torpor-arousal "cycle during hibernation in detail. However, it appears that you are searching for items that have been reported in the past. I feel that the theoretical explanation of the item searched this time is weak. Please improve it. In addition, there are improvements in the entire system, which are described below.

Major comments

The Introduction describes hibernation and the intestinal barrier in detail, both of which involve the small intestine. It is the large intestine that is reported this time, and it is better to include the description of the intestinal barrier in the large intestine. If there are no reports about the large intestine, please include such information. Please also explain why you searched the large intestine this time.

Results provide low-fold and high-fold histological photographs in Figures 1 -5, 7, and 9, respectively. Findings of low magnification are poorly described, and I do not feel the need to include pictures of low magnification. I suppose the authors could remove it. Furthermore, the surface epithelium does not turn upward n figure 3. Please turn them up.

In either result, no actual value was provided. It only shows the increase or decrease compared to each group. Please write the actual value.

My understanding is that DAO activity in serum correlates significantly with DAO activity in small intestinal tissue. Because DAO activity in the small intestine was not examined in this study, it is unclear whether DAO activity in the serum results only from an increase in DAO activity in the large intestine. It would be helpful if you could include that in the discussion.

Materials and Methods

The animal groupings are described in L 706 -707. Please list the number of animals in each group accurately. I do not feel the need to include Figure 14.

Minor comments

About the first appearance of abbreviations

Some abstract descriptions are provided, but this is the first time this report has been published. Please provide an accurate description.

L 134, L 135: HE is the first appearance.

L 136: SA, LT and IBA group.

L 162,163: AB-PAS 

L 254: ET, DAO, and D-LA.

L667; investigatation → investigation

L 170; Is it accurate to use the word recover? Recover is inappropriate because it is not observed over time.

L 933 -944; Do you need these sentences?

Round 2

Reviewer 1 Report

The authors have done a good job revising their manuscript according to my suggestions.  There are two points they requested more information on in their response to my previous comments.  In that document, page 2, Response 4 they asked:   "However, we found a paper that means that they tested the intestinal permeability of the thirteen-lined ground squirrels using only the summer active ground squirrels and the ground squirrels during the interbout arousal period, and I'm not sure if I understand it right, is such an experiment reliable? Because as I understand it, they did not test the intestinal permeability during the torpor period."

Yes, the study described in Carey et al. "Impact of Hibernation on Gut Microbiota and Intestinal Barrier Function in Ground Squirrels" used aroused hibernators to examine intestinal permeability, which is appropriate because at the low body temperature of torpor, there is very little movement of molecules across the intestinal epithelium or anywhere else in the body.  Permeability changes would be therefore most important for the squirrels during an interbout arousal, when molecules can move freely across the gut wall.  Studying permeability in vivo during torpor is difficult, but it can be examined to some extent in vitro as reflected by tissue conductance as shown in Table 1 of Carey, HV, "Seasonal changes in mucosal structure and function in ground squirrel intestine"  Am J. Physiology 1990.  Jejunal tissue conductance in tissues from torpid hibernators tested at low temperature is decreased by over half of the values in tissues from aroused hibernators or summer squirrels.

Top of Page 6: Response 9: you wrote "One thing we are confused about is whether the "body mass differences between the torpor and IBA groups" you are referring to the ratio of colon mass to the weight of the ground squirrels. We would like to know more about this, because, in the future, we will consider quantifying the three data you mentioned to find out more interesting phenomena.".  I was not referring to ratio of colon mass to weight of the ground squirrel, rather simply whether you found any body mass differences between the torpor and IBA groups that you studied.  It is very important to interpretation of all studies to record the body mass of all animals that are used in experiments, but I could not find that information in your manuscript.  

Reviewer 2 Report

I have read the revised paper. Overall, there has been some improvement.

However, my understanding is that when abbreviations are used, the full word should be listed when used for the first time in ABSTRACT and in the text, and then the abbreviation should be presented in parentheses.

Please correct it as such.
